

# Evaluation of soil carbon simulation in CMIP6 Earth System Models

Rebecca M. Varney[1], Sarah E. Chadburn[1], Eleanor J. Burke[2], and Peter M. Cox[1]

[1]*College of Engineering, Mathematics and Physical Sciences, University of Exeter, Laver Building, North Park Road, Exeter, EX4 4QF, UK*
[2]*Met Office Hadley Centre, FitzRoy Road, Exeter, EX1 3PB, UK*

**Correspondence:** Rebecca M. Varney (r.varney@exeter.ac.uk)

**Abstract.** The response of soil carbon represents one of the key uncertainties in future climate change. The ability of Earth System Models (ESMs) to simulate present day soil carbon is therefore vital for reliable projections. In this study the most up-to-date CMIP6 ESMs are evaluated against empirical datasets to assess the ability of each model to simulate soil carbon and related controls: Net Primary Productivity (NPP) and soil carbon turnover time ($\tau_s$). Comparing CMIP6 with CMIP5,

uncertainties in modelled soil carbon remain, particularly the underestimation of northern high latitude soil carbon stocks. There is a robust improvement in the simulation of NPP in CMIP6 compared with CMIP5, however the same improvements are not seen in the simulation of $\tau_s$. These results suggest a greater emphasis is required on improving the representation of below-ground soil processes in future developments of models. These improvements would help reduce the uncertainty of projected carbon release from global soils under climate change and to increase confidence in the carbon budgets associated

with different levels of global warming.

## 1 Introduction

Soil carbon is the Earth's largest terrestrial carbon store, with a magnitude of two to three times the amount of carbon contained within the atmosphere (Jackson et al., 2017). The response of soil carbon to $CO_2$ induced global warming has the potential to provide a significant feedback on climate change, but this feedback is currently poorly known (Friedlingstein et al., 2006;

Gregory et al., 2009; Arora et al., 2013; Friedlingstein et al., 2014; Arora et al., 2020; Song et al., 2021).

    Carbon stored within the atmosphere and global soils is exchanged via carbon fluxes, as part of the global carbon cycle (Ciais et al., 2013). The Earth's terrestrial surface has acted as a carbon sink until now (Pan et al., 2011), but there is a possibility of a switch to a source during the $21^{st}$ century, which would accelerate climate change (Cox et al., 2000; Crowther et al., 2016). Due to the significant quantities of carbon stored in soils globally, understanding and quantifying the potential release of carbon

from soils is vital if the existing Paris Agreement Targets are to be met (UNFCCC, 2015). Earth System Models (ESMs) are complex numerical models which simulate both climate and carbon cycle processes, and are used to make projections of climate change. The most up-to-date ESMs make up the ensemble known as CMIP6, which is the latest phase of the Coupled Model Inter-comparison Project (CMIP) (Eyring et al., 2016), and is used in the most recent Intergovernmental Panel on Climate Change (IPCC) report (AR6) (IPCC, 2021). The response of the carbon cycle to climate change is fundamental to obtaining





accurate future projections, and the relationships between carbon and environmental drivers used in models help to determine this response (Todd-Brown et al., 2013). Therefore, representing present day carbon stores and spatial controls realistically is key for improving the reliability of future projections of climate change.

Present day soil carbon can be approximately broken down into above ground and below ground controls, which influence the spatial distribution of soil carbon stocks (Koven et al., 2015). The above ground control of soil carbon can be considered

as the input flux of carbon into the soil from vegetation. Both the amount of carbon from plant and root litter (known as litter fall), and the fraction of this that is converted to longer-lived soil carbon pools, will influence the storage of soil carbon. Net Primary Productivity (NPP) can be used as a proxy for the litter fall flux, where the fluxes are equal when vegetation is in a steady state. The below ground control of soil carbon can be quantified simply in terms of the soil carbon turnover time ($\tau_s$), which is defined as the time carbon resides in the soil (Koven et al., 2017; Carvalhais et al., 2014). $\tau_s$ can be considered as a

proxy for below ground controls on soil carbon storage (Koven et al., 2015).

In this study, the representation of late $20^{th}$ century soil carbon stores and these related controls, NPP and $\tau_s$, are evaluated in the latest CMIP6 ESMs. Previously, similar studies have been conducted to evaluate soil carbon in the preceding generations of ESMs, for example: Anav et al. (2013) and Todd-Brown et al. (2013) for CMIP5. There are some existing CMIP6 soil carbon related studies, for example: Arora et al. (2020) evaluate carbon-concentration and carbon-climate feedbacks in 1%

$CO_2$ per year forcing simulations, Burke et al. (2020) evaluates the representation of permafrost in models, and Ito et al. (2020) investigate future soil carbon stocks under specific land-use conditions. This study is the first to specifically focus on global and spatial soil carbon and related controls in CMIP6, with a thorough evaluation against empirical datasets and comparison against the preceding CMIP5 ensemble.

The main results of this study are divided into four sections: (1) Soil carbon stocks, (2) Net Primary Productivity, (3) soil

carbon turnover time, and (4) drivers of soil carbon spatial patterns. The first section (1) focuses on soil carbon evaluation on a global and spatial basis, then the second (2) and third (3) sections focus on the evaluation of the related spatial controls of soil carbon, (2) NPP and (3) $\tau_s$. The aim of these results sections is to determine where improvements can be seen in CMIP6 compared with CMIP5, and where improvements are still required based on inconsistencies with empirical datasets. The fourth results section (4) moves from the separate evaluation of modelled soil carbon and related controls, to the simulation of the

relationships between these variables. The aim of this section is to evaluate the modelled relationships against the equivalent relationships derived from the empirical data - simulating realistic relationships between soil carbon and related drivers will help improve the reliability of future climate change projections. Finally, the discussion provides a breakdown of the main results deduced in this study and identifies key areas for future model development.

## 2 Methods

### 2.1 Earth system models


Soil carbon stores and related controls are examined in eleven CMIP6 ESMs (Eyring et al., 2016; Meehl et al., 2014), as shown in Table 1. The ESMs included in this study were chosen due to the availability of the required data in the online repository





at the time of analysis (https://esgf-node.llnl.gov/search/cmip6/). In this study, comparisons are made with the previous CMIP generation of ESMs - CMIP5 (Taylor et al., 2012). Similarly, the CMIP5 models included in this study are listed in Table 2.

Again, models were included if the required data was available online (https://esgf-node.llnl.gov/search/cmip5/).

Tables 1 (CMIP6) and 2 (CMIP5), present information about the included ESMs, specifically more details about the associated Land Surface Model (LSM). It should be noted that there are similarities between some of the Land Surface Models (LSMs) - either advances from earlier models, or even the same LSM within different ESMs. For example, CESM2 and NorESM2-LM both use the Community Land Model version 5 (CLM5) (Arora et al., 2020). For some modelling centres,

both the CMIP5 and CMIP6 versions of the models are included, and in these cases direct comparisons can be made to determine changes from CMIP5 to CMIP6. These generationally related CMIP5 and CMIP6 models are: CanESM2 and CanESM5, CCSM4 and CESM2, GFDL-ESM2G and GDFL-ESM4, IPSL-CM5A-LR and IPSL-CM6A-LR, MIROC-ESM and MIROC-ES2L, MPI-ESM-LR and MPI-ESM1.2-LR, NorESM1-M and NorESM2-LM, and HadGEM2-ES and UKESM1-0-LL, respectively. The models where only either the CMIP5 or CMIP6 version from the modelling centre was included are: BNU-

ESM and GISS-E2-R from CMIP5 and ACCESS-ESM1.5, BCC-CSM2-MR and CNRM-ESM2-1 from CMIP6. A key general change to note is that CMIP6 has more models that include an interactive nitrogen cycle compared with CMIP5: ACCESS-ESM1.5, CESM2, MIROC-ES2L, MPI-ESM1.2-LR, NorESM2-LM and UKESM1-0-LL in CMIP6 compared with CCSM4 and NorESM1-M in CMIP5. (The CMIP5 model BNU-ESM includes carbon-nitrogen interactions, however this process was turned off in CMIP5 simulations (Ji et al., 2014)). Additionally, an increased number of soil carbon pools is seen in some

CMIP6 models (e.g. CLM5 has 29 carbon pools compared with 20 in CLM4). Arora et al. (2020) include a comprehensive overview of the updates seen in the individual CMIP6 models, which is presented in the 'Model descriptions' section of the associated Appendix.

Todd-Brown et al. (2013) include a summary of the temperature and moisture dependencies of soil respiration/decomposition as assumed in the CMIP5 models (see Table 1 of the Todd-Brown et al. (2013) study). The most common representation of the

temperature sensitivity of decomposition is the $Q_{10}$ equation, which is defined by $f(T) = Q_{10}^{(T-T_0)/10}$, where $T$ is temperature and $T_0$ is a reference temperature. With the $Q_{10}$ equation, decomposition increases exponentially with temperature (Davidson and Janssens, 2006). The majority of other models used the Arrhenius equation to represent the temperature sensitivity, where the main difference from the $Q_{10}$ representation is that decomposition levels off at higher temperature levels (Lloyd and Taylor, 1994). Of the remaining models, the GFDL model simulates an increased decomposition with temperature until some optimal

temperature above which it decreases (Shevliakova et al., 2009) - which Todd-Brown et al. (2013) defined as a 'hill' function, and the GISS model implement a linear increase of respiration to temperature up to a maximum value (Del Grosso et al., 2005). The representation of the decomposition sensitivity to soil moisture was found to be to be represented in two ways amongst the CMIP5 models, where either decomposition was assumed to increase monotonically with increasing soil moisture, or less commonly to increase to some optimum moisture level and then decrease (again described as a 'hill' function by Todd-Brown

et al. (2013)). In this study we note that the representation of temperature and moisture functions remain similar from CMIP5 from CMIP6. The $Q_{10}$ equation remains the most common representation of soil temperature sensitivity in models, followed





by the Arrhenius equation and then 'hill' functions. Similarly, the most common representation of the sensitivity of soil to moisture in CMIP6 is a monotonically increasing function, followed by 'hill' functions of various sorts.

## 2.2 Defining soil carbon variables

CMIP defines common output variables (Meehl et al., 2000), which allows for consistent comparison between the models, and for cleaner evaluation of models to observational data. These common output variables also allow for consistent comparison between model generations, in this case between CMIP6 and CMIP5. This study focuses on evaluation of near present day soil carbon and related controls. Therefore the results presented in this study use the CMIP standard historical simulation (CMIP scenario *historical*), for both the CMIP6 and CMIP5 analysis. The historical simulation runs from 1850-2015 in CMIP6 and

from 1850-2005 in CMIP5, where the selected dates for each variable (stated below) were chosen to allow for consistent comparison between CMIP5 and CMIP6, and to best match the modelled data to the observational data.

To evaluate soil carbon, this study uses 'Soil Carbon' (CMIP variable *cSoil*) which represents the carbon stored in soils, and where applicable 'Litter Carbon' (CMIP variable *cLitter*) which represents carbon stored in the vegetation litter. Total soil carbon, $C_s$, is defined to be the sum of these soil carbon and litter carbon variables (*cSoil + cLitter*), where applicable. This

allows for a more consistent comparison between the models and between the models and empirical data, due to differences in how soil carbon and litter carbon are simulated (Todd-Brown et al., 2013; Arora et al., 2020). Where for models that do not report a separate litter carbon pool, the total soil carbon is taken to be simply the *cSoil* variable. Modelled $C_s$ is time averaged between the years 1950 to 2000 of the historical simulation, and is considered spatially (units of kg m$^2$), and as global totals (units of PgC), where global totals are calculated as an area weighted sum (using the model land surface fraction,

CMIP variable *sftlf*) and divided by $1 \times 10^{12}$). To calculate northern latitude totals, a sum between the latitudes 60° N and 90° N is considered.

The CMIP6 models CESM2 and NorESM2-LM have two different variables to represent soil carbon: CMIP variable *cSoil*, which represents the full vertical soil profile, and CMIP variable *cSoilAbove1m*, which represents soil carbon in the top 1m of soil. This is due to the representation of vertically resolved soil carbon in these models, which means there are separate

carbon pools in the model that represent different soil depths (Lawrence et al., 2019). The CMIP variable *cSoilAbove1m* is used throughout this study to represent soil carbon for the models CESM2 and NorESM2-LM, unless otherwise stated. The use of this variable is to enable a more consistent comparison with both the other CMIP6 models and the CMIP5 models. Therefore, an assumption of a 1m depth of soil for modelled soil carbon allows for the fairest evaluation, and evaluation is considered against empirical datasets down to a depth of 1m (see below). However, comparisons with the *cSoil* variable for

both CESM2 and NorESM2-LM are included in Tables 4 and 6 of the Results.

In order to obtain a clean separation between above-ground and below-ground drivers of soil carbon variations, a quasi-equilibrium approximation is made. We begin with the definition of the effective soil carbon turnover time ($\tau_s$) (Varney et al., 2020; Koven et al., 2017; Carvalhais et al., 2014), which represents the average time carbon resides in the soil:

$$\tau_s = C_s/R_h \tag{1}$$





where $R_h$ is the output flux of carbon from the soil, known as the heterotrophic respiration. This definition of the turnover time implicitly neglects other processes that may release soil carbon, but which are not yet routinely included in ESMs (e.g. peat fires or dissolved organic carbon fluxes).

In an unperturbed steady-state (i.e. neglecting disturbances from land-use change, fires, insect outbreaks etc.), there is no net exchange of carbon between land and atmosphere, and $R_h$ is equal to the Net Primary Productivity (NPP, $\Pi_N$). In the

contemporary period, the difference between $\Pi_N$ and $R_h$ represents the Net Ecosystem Productivity (NEP, $\approx 3$ PgC yr$^{-1}$), which is small compared to the $\Pi_N$ and $R_h$ fluxes ($\approx 60$ PgC yr$^{-1}$). Therefore the present day soil carbon can be approximated by:

$$C_s \approx \Pi_N \, \tau_s \tag{2}$$

which gives a clean separation of soil carbon variation into the above (NPP) and below ($\tau_s$) ground drivers of soil carbon spatial

patterns.

To evaluate these soil carbon controls on $C_{s,eq}$, NPP and $\tau_s$ are evaluated separately. This study uses modelled 'Net Primary Productivity' (CMIP variable *npp*), which is defined as the mass flux of carbon out of the atmosphere due to NPP on land. NPP is also considered spatially (kg m$^2$ yr$^{-1}$), and as an area weighted global total flux (PgC yr$^{-1}$). By definition $\tau_s$ is defined by Eq. 1, so $\tau_s$ is calculated by soil carbon (as defined above) divided by $R_h$. $R_h$, 'Heterotrophic Respiration' (CMIP variable *rh*),

is defined as the mass flux of carbon into the atmosphere due to heterotrophic respiration on land, primarily due to the microbial respiration that occurs in the soil, and where the units of $R_h$ are the same as that of NPP. The carbon fluxes (NPP and $R_h$) are time averaged over the period 1995-2005 for consistency between the CMIP generations and to match the empirical datasets. $\tau_s$ can be considered on a spatial level, or as an effective global $\tau_s$, which is defined as average $\tau_s$=mean($C_s$)/mean($R_h$) (where the mean represents an area weighted global average). The advantage of defining an effective global $\tau_s$ is that it is not dominated

by large spatial outlying values. Using either method, the units for $\tau_s$ are in years by definition.

The relationships of soil carbon, $C_s$, NPP and $\tau_s$, with both temperature and soil moisture are also considered. For temperature, the variable 'near surface air temperature' (CMIP variable *tas*), representing atmospheric temperature at the surface is considered, where the dates 1995-2005 where chosen to be consistent with the carbon fluxes. The variable for atmospheric temperature is considered opposed to soil temperature as equivalent global observational datasets are required for the analy-

sis. For soil moisture, the variable 'Moisture in Upper Portion of Soil Column' (CMIP variable *mrsos*), which is defined as the mass content of water in the soil layer in the upper portion of the soil (0cm-10cm depth) is considered, where the dates 1978-2000 were considered to match the empirical data. The standard output *mrsos* is in units of kg m$^2$, however in this study a volumetric soil moisture, referred to as $\theta$, is used to allow for consistent comparison with the benchmark data. $\theta$ is calculated as *mrsos* divided by the depth of the soil layer in mm, which in this case is $\theta$ = mrsos/100. The variable *mrsos* for soil moisture

was considered opposed to the full soil column moisture (CMIP variable *mrso*) as this better matched the available empirical dataset for soil moisture.



## 2.3 Empirical datasets

### 2.3.1 Soil carbon

Observational soil carbon, $C_s$, to a depth of 1m, was obtained by combining the empirical Harmonized World Soils Database
(HWSD) (FAO and ISRIC, 2012) and Northern Circumpolar Soil Carbon Database (NCSCD) (Hugelius et al., 2013) soil carbon
datasets, where NCSCD was used where overlap of the datasets occurs. This is a commonly used method when considering
empirical soil carbon and has been previously used in multiple studies, such as: Varney et al. (2020), Koven et al. (2017), and
Todd-Brown et al. (2013). This dataset is referred to here as the 'Benchmark dataset'.

We use the 95% confidence intervals given by Todd-Brown et al. (2013), to derive standard deviations about the global
mean soil carbon. To do this, the constructed 95% confidence intervals where used to calculate upper and lower bounds around
the mean value. Then assuming the data is normally distributed, these derived 95% confidence intervals were halved to obtain
confidence intervals equivalent to a standard deviation error on the mean ($1412 \pm 215$ PgC). The uncertainty analysis completed
in Todd-Brown et al. (2013) is used for the benchmark soil carbon dataset as no quantitative uncertainty has been previously
or since defined for the HWSD and NCSCD datasets (Anav et al., 2013).

Additionally, the benchmark dataset was compared with empirical estimates found in the literature to improve the robustness
and reliability of the evaluation. Todd-Brown et al. (2013) find that this derived uncertainty is consistent with other empirical
estimates of global soil carbon; for example, 1576 PgC in Eswaran et al. (1993), 1220 PgC in Sombroek et al. (1993), and 1502
PgC in Jobbágy and Jackson (2000). This study further compares with empirical estimates of 1395 PgC in Post et al. (1982)
and 1515 PgC in Raich and Schlesinger (1992). These empirical estimates are with one standard deviations of the global mean
soil carbon given by the benchmark dataset (Table 3).

These additional datasets include: (1) the World Inventory of Soil property Estimates (WISE30sec) dataset down to a depth
of 2m (Batjes, 2016), which includes a given standard deviation on the global total soil carbon consistent with our derived
benchmark uncertainty, (2) the named 'S2017' from Sanderman et al. (2017) soil carbon estimate (1m and 2m), which uses a
data-driven statistical model and the History Database of the Global Environment (HYDE) land use data, (3) the Global Soil
Dataset for use in Earth System Models (GSDE), which provides a estimates for observational soil carbon down to a depth
of up to 2.3m (Shangguan et al., 2014), and (4) the Global Gridded Surfaces of Selected Soil Characteristics (IGBP-DIS)
estimate of soil carbon to a depth of 1m, derived by the Oak Ridge National Laboratory Distributed Active Archive Centre
(ORNL DAAC) (IGBP, 2000). These datasets were combined to obtain a mean estimate for observational soil carbon down
to a depth of 1m, where a global total soil carbon value of $1560 \pm 214$ PgC was found. This estimate is consistent with
our benchmark dataset estimate, and further improves the confidence in our benchmark soil carbon estimate. Furthermore, the
spatial correlation coefficients between these additional datasets and our benchmark dataset are considered, where the following
values correspond to the above datasets: (1) 0.554, (2) 0.625, (3) 0.482, and (4) 0.622. Map plots comparing the empirical soil
carbon datasets are shown in Fig. A1. The estimate for northern latitude total soil carbon has greater uncertainties associated
with it, where the standard deviation deduced by combining the empirical datasets is 83 PgC. To account for this increased



uncertainty, the deduced standard deviation of 83 PgC is used on the benchmark soil carbon throughout this study, opposed to the 61 PgC derived using the Todd-Brown et al. (2013) uncertainty analysis.

### 2.3.2  Carbon fluxes

To estimate a benchmark Net Primary Production (NPP), the commonly used MODIS NPP (2000-2010) dataset (Zhao et al., 2005) is used. The MODIS NPP dataset does not have associated uncertainty estimates, so this study estimates a standard
deviation error on benchmark NPP as derived by Ito (2011). The MODIS NPP dataset is found to be consistent with 251 empirical present day estimates of NPP found in the literature, which Ito (2011) used to estimate a global value of $56.2 \pm 14.3$ PgC yr$^{-1}$ (compared with a derived MODIS mean value of 56.6 yr$^{-1}$). Moreover, due to the limited choice of observational derived NPP datasets (Harper et al., 2018), models can be further evaluated against using a benchmark dataset for Heterotrophic respiration ($R_h$), where $R_h$ is estimated using the CARDAMOM (2001–2010) heterotrophic respiration dataset (Bloom et al.,
2015). The empirical CARDAMOM $R_h$ has associated estimates of error, which were used to derive a standard deviation uncertainty on the empirical average $R_h$ ($51.7 \pm 21.8$ PgC yr$^{-1}$). This study includes map plots comparing the two empirical datasets, which is shown in Fig. A2. Global totals for $R_h$ are also considered for comparison against NPP, where the CMIP6 and CMIP5 values are also shown in Appendix Tables A1 and A2, respectively.

### 2.3.3  Soil carbon turnover time

To estimate a benchmark soil carbon turnover time ($\tau_s$), the estimates of observational soil carbon are divided by an estimate of heterotrophic respiration ($R_h$) (see above). To estimate an uncertainty on effective global $\tau_s$, this study derived upper ($\tau_s^+$) and lower ($\tau_s^-$) bounds based on the derived $C_s$ and $R_h$ uncertainty estimates. The upper bound was calculated using the following: $\tau_s^+ = C_s^+ / R_h^-$, where $C_s^+$ is equal to the mean soil carbon plus one standard deviation and $R_h^-$ is equal to the mean heterotrophic respiration minus one standard deviation. The lower bound was calculated using the following: $\tau_s^- = $
$C_s^- / R_h^+$, where similarly $C_s^-$ is equal to the mean soil carbon minus one standard deviation and $R_h^+$ is equal to the mean heterotrophic respiration plus one standard deviation. This method gives a large uncertainty bound around the derived mean estimate ($27.0_{-11}^{+27}$ yr), so the benchmark data is further compared to empirical estimates. Raich and Schlesinger (1992) derive an estimate of mean soil carbon turnover of 32 yr, using estimates for mean soil carbon pools and mean soil respiration rates. More recently, Carvalhais et al. (2014) derive an estimate for the mean global ecosystem carbon turnover time of $23_{-4}^{+7}$, which
is a spatially explicit and observation based estimate. Ito et al. (2020) derived an observational uncertainty range on soil carbon turnover time of 18.5 to 45.8 years, which was derived using similar empirical estimates found in the literature. These estimates give more certainty on the values closer to the derived empirical mean value for $\tau_s$.

### 2.3.4  Soil moisture and air temperature

To estimate soil moisture ($\theta$), the Copernicus Climate Change Service (C3S) 'Soil moisture gridded data from 1978 to present'
dataset (published 2018-10-25) is used, where the years 1978 to 2000 are considered. This dataset is based on the ESA Climate



Change Initiative soil moisture, and estimates global surface soil moisture from a large set of satellite sensors (Copernicus Climate Change Service, 2021; Liu et al., 2011, 2012; Wagner et al., 2012; Gruber et al., 2017; Dorigo et al., 2017). The WFDEI Meteorological Forcing dataset is used to represent observational air temperatures (1995-2005) (Weedon et al., 2014), where dates are choosen to allow for consistency between CMIP generations. This study includes no uncertainty analysis on the soil moisture and air temperature empirical datasets as these datasets are only used to evaluate spatial correlations with modelled data and not to evaluate soil moisture and air temperature in the models.

## 2.4 Regridding

To allow direct comparisons between the empirical data and model output data, the model data was regridded to match the observational grid. In this case, the observational grid is a 0.5° by 0.5° resolution, 720 longitude and 360 latitude grid. This was done using Iris - the community-driven Python package for analysing and visualising Earth science data (Met Office, 2010 - 2013). The regidding method assumed conservation of mass and used linear extrapolation, where extrapolation points will be calculated by extending the gradient of the closest two points. Moreover, model land masks are used to calculate the fraction of land in each coastal grid cell.

## 2.5 Statistical analysis

It is difficult to evaluate the spatial distributions of modelled soil carbon and related spatial controls against empirical data with a single metric, so the evaluation for both CMIP6 and CMIP5 involves multiple methods. These include: coefficients of variation, spatial standard deviations, spatial Pearson correlation coefficients and Root Mean Square Errors (RMSEs). These methods can be combined to give a more thorough evaluation of spatial soil carbon and associated controls in the CMIP6 models compared to the previous generation of CMIP5 models.

The coefficient of variation is defined as the ratio of the ensemble standard deviation (std) to the ensemble mean in each grid cell. This is used to show the amount of variability amongst the models in the ensemble scaled to the size of the ensemble mean, so represents the variability spatially in the ensemble and shows how much variation is present across the ensemble in specific regions. It is presented as hatching on map figures, where shaded 'hatched' regions show regions of high variability within the ensemble. These regions show areas where there is disagreement in the ensemble as there is large spread compared with the mean, and is defined as where $std/mean > 0.75$. The regions where spatial $C_s < 5$ kg m$^2$ are discounted as the low values of soil carbon discounts the significance of disagreement in these regions.

The spatial standard deviation is a measure of the spread in the data across the globe compared to the mean value. Pearson correlation coefficients (r-values) are used as a spatial measure of the linear correlation between the empirical and modelled data, where a high r-value (near 1 or -1) represents a high correlation in the data and a low r-value (near 0) represents a negligible correlation. Root Mean Square Error (RMSE) is used as an absolute measure of the difference between the modelled data and empirical data, where the lower the value the lower the difference error. The RMSE can be considered as the standard deviation of the difference, and it is a measure to show the deviation of the modelled data in relation to the empirical data. This statistical data: spatial standard deviations, Pearson correlation coefficients, and RMSEs, can be presented using a Taylor



diagram. A Taylor diagram is a mathematical graph used to indicate the performance of a model compared with a benchmark,
which in this case is the empirical datasets (Taylor, 2001).

## 3 Results

### 3.1 Soil carbon stocks: *northern latitude underestimations remain in CMIP6*

#### 3.1.1 Global total evaluation

Global total soil carbon (in the top 1m of soil) is shown to vary amongst the ESMs in CMIP6, with a range of 1294 PgC between
the models with the lowest and the highest values (Table 4). The global total soil carbon for two (CanESM5 and MIROC-ES2L)
out of the eleven CMIP6 models falls within the benchmark soil carbon uncertainty range, 1197 - 1627 PgC (mean ± stand
deviation). The models with the largest global total soil carbon are CNRM-ESM2-1 (1810 PgC), BCC-CSM2-MR (1770 PgC),
and UKESM1-0-LL (1760 PgC), values greater than the benchmark dataset but not the additional empirical datasets (Table 3).
The models GFDL-ESM4 (516 PgC) and IPSL-CM6A-LR (639 PgC) have the lowest global total soil carbon values in the
ensemble, with global totals significantly lower (approximately 50% less) than the global totals seen in empirical data. It is
noted that if the full soil carbon profile is considered for CESM2 and NorESM2-LM opposed to a depth of 1m, the global total
soil carbon values are increased to 1870 PgC from 991 PgC in CESM2, and to 2430 PgC from 969 PgC in NorESM2-LM.

  The ensemble mean global total soil carbon is found to have reduced in CMIP6 from CMIP5. Table 4 includes the CMIP6
ensemble mean global total soil carbon, where a total of 1206 ± 445 PgC is deduced, using regridded model resolutions (see
methods). It is noted that Ito (2011) state a CMIP6 ensemble of 1553 ± 672 PgC, however the full soil carbon profile is consid-
ered for CESM2 and NorESM2-LM, opposed to a depth of 1m considered in this study. Table 5 shows the CMIP5 equivalent
soil carbon values, where an ensemble mean global soil carbon value of 1480 ± 810 PgC is deduced, using equivalent dates in
the historical simulation (1950-2000). Todd-Brown et al. (2013) state an ensemble mean soil carbon value of 1520 ± 770 PgC
in CMIP5, however the Todd-Brown et al. (2013) study includes the models BCC-CSM1.1, CESM1-CAM5 and INM-CM4,
which are missing from the analysis in this study due to data availability. Anav et al. (2013) present a CMIP5 ensemble mean
soil carbon value of 1502 ± 798 PgC, but this calculation includes multiple model versions (for example, LR and MR) from
the same modelling centre in their ensemble.

  Despite a reduction in ensemble mean global total soil carbon in CMIP6 compared with CMIP5, the CMIP6 ensemble value
remains within the benchmark uncertainty range. However, a significant reduction is seen in the associated standard deviation
of the ensemble mean global totals (± 445 PgC in CMIP6 from ± 810 PgC in CMIP5). Moreover, a reduced range of global
total values is seen in CMIP6 compared with CMIP5, where a range of 1294 PgC is seen in CMIP6 opposed to 2493 PgC in
CMIP5. This suggests that although a significant range in global soil carbon still exists amongst the CMIP6 ESMs, there is an
improved consistency between the models seen in CMIP6 compared with the models in CMIP5. It is found from comparing
the previous generation models in CMIP5 with the updated CMIP6 equivalent, that multiple models in CMIP6 have lower
quantities of soil carbon than in CMIP5, such as: GFDL-ESM4 from GFDL-ESM2G, IPSL-CM6A-LR from IPSL-CM5A-LR,



MIROC-ES2L from MIRCO-ESM and MPI-ESM1.2-LR from MPI-ESM-LR. For example, the CMIP5 model MPI-ESM-LR is reported to have the largest soil carbon magnitude amongst the CMIP5 models, with a global total of 3000 PgC (Table 5), whereas the updated CMIP6 model MPI-ESM1.2-LR has a reduced global total soil carbon value of 970 PgC, amongst the lowest values reported in CMIP6 and below observational derived range (Table 4). Conversely, these reductions are negated in

the ensemble mean by the remaining models which have greater quantities of soil carbon in CMIP6 compared to their CMIP5 equivalent, such as CanESM5 from CanESM2, CESM2 from CCSM4, NorESM2-LM from NorESM1-M and UKESM1-0-LL from HadGEM2-ES. For example, the CMIP5 model NorESM1-M is amongst the lowest soil carbon values presented in this ensemble at 538 PgC (Table 5), whereas the updated CMIP6 model NorESM2-LM has an increased global total of 969 PgC (down to 1m) (Table 1).

### 3.1.2 Northern latitude total evaluation

Northern latitude soil carbon (down to a depth of 1m, and where northern latitudes defined as 60° N - 90° N) is found to be underestimated in CMIP6, with eight out of the eleven CMIP6 models having lower northern latitude soil carbon values than the derived observational range (Table 4). Two out of eleven CMIP6 models (CNRM-ESM2-1 and MIROC-ES2L) have northern latitude totals that fall within the uncertainty range derived from the benchmark data, 318 - 484 PgC (mean ± stand deviation).

The CMIP6 models with the greatest northern latitude total soil carbon are BCC-CSM2-MR (575 PgC), CNRM-ESM2-1 (440 PgC), and MIROC-ES2L (347 PgC). The CMIP6 models with the lowest northern latitude soil carbon are IPSL-CM6A-LR (66 PgC), ACCESS-ESM1.5 (151), GFDL-ESM4 (163 PgC), MPI-ESM1.2-LR (175 PgC) and UKESM1-0-LL (194), values significantly lower than the totals seen in empirical data.

The northern latitude soil carbon total was also underestimated in CMIP5, with six out of the ten CMIP5 models estimating
northern latitude totals lower then the empirical estimates (Table 5). The ensemble mean total northern latitude soil carbon is lower in CMIP6 (266 ± 139 PgC seen in Table 4) than in CMIP5 (318 ± 246 PgC seen in Table 5), which is consistent with the global total results, however both the CMIP5 and CMIP6 mean values fall below the benchmark range. Similarly, as with global soil carbon, a smaller standard deviation on the mean is found for CMIP6 compared with CMIP5. Moreover, there is a reduced range in simulated northern latitude total values amongst the CMIP6 models, where despite a large range
seen (66 to 575 PgC), an even greater range is seen in CMIP5 (28.1 to 742 PgC). Moreover, improvements are seen amongst models from CMIP5 to CMIP6. For example, the CMIP5 model NorESM1-M had a northern latitude total soil carbon value of 31.0 PgC, which is significantly lower than what is expected based on the benchmark dataset (Table 5). However, the updated CMIP6 version of this model, NorESM2-LM, has a northern latitude total soil carbon value of 300 PgC, which is much more in line with the expected observational values (Table 4). An improved representation of northern latitude soil carbon is also
seen CESM2 (compared with CCSM4), which has the same land surface model as NorESM2-LM (CLM5 (Lawrence et al., 2019)).

The CMIP6 models with the lowest global total values for soil carbon do not always correspond with the lowest northern latitude values for soil carbon. For example, UKESM1-0-LL global total soil carbon is amongst the highest global totals seen in CMIP6, however low quantities of soil carbon are seen in the northern latitudes (approximately 10% of the global total).





Conversely, BCC-CSM2-MR, CESM2, GFDL-ESM4, and NorESM2-LM have approximately 30% of their global total stocks in the northern latitude region, which is consistent with the ratio seen in the benchmark dataset. This result suggests that representing global total soil carbon stocks consistent with the benchmark soil carbon, does not imply the consistency in the representation of northern latitude soil carbon stocks, and these should be evaluated separately. However, the large uncertainties associated with the empirical datasets for the northern latitudes are noted (Table 3).

**3.1.3  Spatial evaluation**

A lack of consistency in the simulation of soil carbon was found amongst the CMIP5 models, which can be seen in Fig. 1(a), where differences between the empirical and modelled data is shown. Northern latitude soil carbon was found to be underestimated in CMIP5, where areas of blue can be seen in the northern latitudes of the CMIP5 soil carbon map in Fig. 1(a). This underestimation of CMIP5 northern latitude soil carbon is accompanied by significant overestimations seen in mid-

latitude soil carbon. Specifically, large quantities of soil carbon which are inconsistent with our benchmark dataset can be seen in the mid-latitude regions in the following CMIP5 models: CanESM2, GFDL-ESM2G, GISS-E2-R, MIROC-ESM, and MPI-ESM-LR, and less significant overestimations are seen in HadGEM2-ES and IPSL-CM5A-LR (Fig. A3). Systematic errors remain in the CMIP6 models, however there are some improvements seen in the spatial simulation of soil carbon from CMIP5. Soil carbon is still underestimated in the northern latitudes, where the areas of blue still remain the northern

latitudes of the CMIP6 soil carbon map in Fig. 1(a). However, regions of overestimations in the northern latitudes are also seen amongst the CMIP6 models in BCC-CSM2-MR, CESM2, CNRM-ESM2-1, and NorESM2-LM (Fig. 2), but it is noted that this representation might be more consistent with observations if a dataset including deeper soil carbon stocks was considered. CMIP6 shows improvements in the representation of mid-latitude soil carbon, where less of an overestimation is seen in CMIP6 compared with CMIP5 (Fig. 1(a)). This overestimation can still be seen in four of the eleven CMIP6 models: ACCESS-

ESM1.5, CanESM2, MIROC-ES2L and UKESM1-0-LL, however the overestimations in CMIP6 are less inconsistent than when compared with CMIP5 and the number of models showing this limitation in CMIP6 has been reduced (Fig. 2).

Despite the differences seen in the spatial representation of soil carbon between the individual models in CMIP6, the ensemble mean has more areas of agreement within the ensemble compared to the ensemble mean in CMIP5. This can be seen in Fig. 3(a), where there is less hatching (where hatched shaded areas represent regions of low agreement amongst the models in the

ensemble, see methods) in the CMIP6 map compared with the CMIP5 map. Specifically, ensemble mean soil carbon in CMIP6 has more areas of agreement in the mid-latitude region compared with the CMIP5 ensemble mean, where significant areas of disagreement are seen. This disagreement is likely due to the overestimation which exists in some of the CMIP5 models (Fig. A3). Also, a reduction in the area of disagreement is seen in the northern latitudes in CMIP6 compared with CMIP5, however this remains the region where the most disagreement exists across the generations. It is noted that this is a measure of

agreement within the ensemble, and not between the models and empirical data.

The inconsistency of the simulation of spatial soil carbon in CMIP6 is further evaluated using the spatial standard deviations, the spatial Pearson correlation coefficients and Root Mean Square Errors (RMSEs) (see Methods). The Taylor Diagram (Fig. 4(a)) presents all three statistical assessments. The spatial standard deviation for soil carbon is shown on the radial axis between





standard x and y axes in Fig. 4(a). The range of spatial standard deviations amongst the CMIP6 models sees a slight reduction

from the range amongst the CMIP5 models, though significant differences remain. The CMIP6 models CNRM-ESM2-1, MIROC-ES2L and UKESM1-0-LL best match the spatial standard deviation derived from the benchmark dataset (Tables 4 and 5). It is found that the spatial representation of modelled soil carbon in CMIP6 is poorly correlated to the empirical soil carbon, where the CMIP6 ensemble spatial correlation coefficient with the empirical data is found to be 0.250. The spatial correlation coefficients between the individual CMIP6 and CMIP5 models with the empirical data can also be seen in Fig.

4(a), where the low spatial correlation coefficients are shown by the curved correlation axis. The lowest spatial correlation coefficients amongst the CMIP6 models were r-values of 0.104 in IPSL-CM6A-LR and 0.115 in UKESM1-0-LL. The CMIP6 model that was the most spatially consistent with the empirical data is CNRM-ESM2-1, with an r-value of 0.630. The CMIP6 ensemble sees a slight reduction in the RMSE compared to the CMIP5 ensemble, suggesting a slight improvement (Fig. 5(a)). Significant improvements in the RMSE are seen in MIROC-ES2L from MIROC-ESM and MPI-ESM1.2-LR from MPI-ESM-

LR. These results suggest small improvements in the simulation of soil carbon across this CMIP generation, however the low spatial correlation coefficients and variable RMSEs seen across the models in CMIP6 suggest inconsistencies with the benchmark data remains.

### 3.2    Net Primary Productivity: *improved in CMIP6 relative to CMIP5*

#### 3.2.1    Global total evaluation

Global total NPP amongst the CMIP6 models is consistent with the benchmark dataset (Table 6), where the CMIP6 ensemble mean for NPP is approximately 95% of the benchmark mean. The CMIP6 ensemble mean global total NPP (53.0 $\pm$ 9.39 PgC yr$^{-1}$) is found to be slightly lower than the derived mean benchmark value, however it is comfortably within the observational uncertainty range (56.6 $\pm$ 14.3 PgC yr$^{-1}$). The equivalent values for the CMIP5 models can be seen in Table 7, where the CMIP5 ensemble total is also found to be within the observational uncertainty range (56.3 $\pm$ 15.4 PgC yr$^{-1}$). The standard

deviation surrounding the CMIP5 ensemble mean is greater than in CMIP6. This reduced uncertainty in CMIP6 is because several of the models have a simulated global total NPP that more closely matches the benchmark NPP global total value compared with the previous CMIP5 generation. For example, GFDL-ESM4 from GFDL-ESM2G, IPSL-CM6A-LR from IPSL-CM5A-LR, MIROC-ES2L from MIROC-ESM, MPI-ESM1.2-LR from MPI-ESM1-M, and UKESM1-0-LL from HadGEM2-ES. The majority of CMIP6 models see a reduction in NPP from the CMIP5 equivalent model, which in general reduces the

overestimation of NPP that was seen in the CMIP5 models (Table 7 and 6). However, is was not the case for CanESM5 from CanESM2 which sees an increase in the magnitude of NPP from CMIP5 to CMIP6, resulting in a consequent overestimation compared to the benchmark data. A reduced range of modelled global total NPP values is also seen in CMIP6 from CMIP5, where the range is reduced from 48.5 PgC yr$^{-1}$ in CMIP5 to 32.7 PgC yr$^{-1}$ in CMIP6. These results suggest that overall the representation of carbon fluxes in CMIP6 ESMs is more consistent than in CMIP5.





### 3.2.2  Spatial evaluation


Modelled NPP in CMIP6 is spatially more consistent with empirical data than CMIP5. This can be seen in Fig. 1(b), where the difference between the modelled and benchmark NPP is shown for both CMIP5 and CMIP6. It can be seen in the CMIP5 map that NPP is overestimated in the tropical regions, specifically in Africa and South East Asia, and the equivalent CMIP6 difference map shows a clear reduction in this overestimation. This tropical overestimation of NPP prominent in CMIP5 (Fig. A4), is still seen in the CMIP6 models CanESM5, MPI-ESM1.2-LR and UKESM1-0-LL. However, this is not seen in the CMIP6 ensemble mean as it is likley negated by underestimations seen in CESM2, CNRM-ESM2-1, and NorESM2-LM (Fig. 6). CMIP6 also sees more consistency with the benchmark dataset in the northern and mid-latitudes compared with CMIP5, where more white areas are seen in the CMIP6 map in Fig. 1(b). An underestimation of NPP is seen in both CMIP5 and CMIP6 on the west side of South America, though unusually high NPP is seen in this region in the MODIS NPP dataset (Fig. A2). Moreover, greater agreement amongst the models within CMIP6 is seen compared the models in CMIP5. This can be seen in Fig. 3(b), where less hatching representing areas of disagreement within the ensemble is seen in the CMIP6 compared with CMIP5. Specifically, CMIP6 sees less hatching in the northern latitudes, the Middle East and South East Europe, as well as regions in South America, South Africa and Australia.

The improved empirical consistency of modelled NPP in CMIP6 is also found when further evaluated using the same spatial metrics as with soil carbon. Despite a small range remaining in the spatial standard deviations amongst the CMIP6 models (shown by the radial axis in Fig. 4(b)), robust improvements in the spatial correlation coefficients (shown by the curved axis in Fig. 4(b)) and RMSEs are seen across the ensemble compared with CMIP5 (Fig. 5(b)). Notable improvements in the representation of NPP are seen in GFDL-ESM4 compared with GFDL-ESM2G, IPSL-CM6A-LR compared with IPSL-CM5A-LR, and UKESM1-0-LL compared with HadGEM2-ES, with reduced RMSEs seen in each updated model. A general improvement in the spatial correlation coefficients is seen across all the CMIP6 models, where the circle markers (CMIP6 models) in Fig. 4(b), have higher correlation values than the cross markers (CMIP5 models). The general improvement has resulted in the CMIP6 ensemble correlation coefficient (0.836) being greater compared with the equivalent CMIP5 value (0.711). The lowest correlations between modelled and observed NPP amongst the CMIP5 models are GISS-E2-R (0.274) and CanESM2 (0.469). The updated version CanESM5 remains the lowest correlation seen in CMIP6 (0.655), however an improvement in the correlation is seen. The updated version of the GISS model is not included in the CMIP6 ensemble considered in this study, which could be a reason for the increased ensemble mean correlation. However, this effect does not take away from the improvements seen across the CMIP6 models. HadGEM2-ES (0.764) and MPI-ESM-LR (0.764) were the CMIP5 models with the highest correlation to the benchmark NPP, and the updated CMIP6 equivalents of these models remain the models with the greatest correlations, but again improvements in the correlations are seen (0.816 in UKESM1-0-LL and 0.785 MPI-ESM1.2-LR).



### 3.3 Soil carbon turnover time: *no major improvements in CMIP6 compared to CMIP5*

#### 3.3.1 Global evaluation

There are minor improvements seen in the simulated effective global $\tau_s$ amongst select CMIP6 models (Table 6) compared with CMIP5 (Table 7). The ensemble mean effective global $\tau_s$ was overestimated in CMIP5 (37.8 ± 19.7 yr) when compared

with the derived mean $\tau_s$ using the benchmark datasets ($27.0^{+27}_{-11}$ yr), which is reduced to a less significant underestimation in CMIP6 (23.3 ± 8.59 yr). However, both the CMIP5 and CMIP6 estimates fall within the observational uncertainty range. The associated ensemble uncertainty on effective mean $\tau_s$ is less in CMIP6 compared with CMIP5, with a ensemble standard deviation of approximately 50% less. A significant range is seen in the effective global $\tau_s$ values amongst the CMIP5 models, with 5 fold difference between the lowest and the highest values (Table 7). This range is mostly due to large overestimations

seen amongst the CMIP5 models, for example in CanESM2, GFDL-ESM2G and MIROC-ESM. A reduced range is seen in amongst the models in CMIP6, however a 4 fold range still exists between the lowest and the highest values (Table 6). This reduced range is partly due to reductions in the effective global $\tau_s$ values in CMIP6 models compared to the equivalent model in CMIP5, specifically, CanESM5 from CanESM2, GFDL-ESM4 from GFDL-ESM2G, MIROC-ES2L from MIROC-ESM, and MPI-ESM1.2-LR from MPI-ESM-LR. Though overestimations do remain in CMIP6, for example in CNRM-ESM2-1,

where the slowest effective turnover time was seen. Moreover, the range is also reduced due to improvements seen in models which underestimated $\tau_s$ in CMIP5, such as UKESM1-0-LL from HadGEM2-ES and CESM2 from CCSM4.

#### 3.3.2 Spatial evaluation

The comparison of spatial soil carbon turnover times ($\tau_s$) in CMIP6 with CMIP5 has more varied results than comparing simulated NPP. The CMIP5 ensemble showed an underestimation of $\tau_s$ in the northern latitudes, which is replaced with an

overestimation of $\tau_s$ in CMIP6 when compared to the benchmark data (Fig. 1(c)). This northern latitude overestimation in the CMIP6 ensemble is a result of the overestimations of $\tau_s$ in CESM2 and NorESM2-LM (Fig. 7), which dominate in the CMIP6 ensemble mean. It is noted that this result may differ if deeper soil carbon stocks were considered. The northern latitude underestimation of $\tau_s$ is still seen within the CMIP6 models, such as CanESM5, CNRM-ESM2-1, GFDL-ESM4, IPSL-CM6A-LR, MIROC-ES2L, MPI-ESM1.2-LR, and UKESM1-0-LL (Fig. 7). An overestimation of mid-latitude $\tau_s$ was

seen in the CMIP5 models MIROC-ESM and MPI-ESM-LR (Fig. A5), which is no longer seen in the updated CMIP6 models MIROC-ES2L and MPI-ESM1-2-LR, respectively. However, an overestimation of mid-latitude $\tau_s$ is seen in CMIP6 models BCC-CSM2-MR, CNRM-ESM2-1 and UKESM1-0-LL (Fig. 7). The uncertainty in simulated northern latitude $\tau_s$ is also apparent in Fig. 3(c), where the hatching shows the lack of agreement within the CMIP6 ensemble in this region. However, more agreement within the CMIP6 ensemble is seen in the same figure in the mid-latitudes and tropical regions compared with

CMIP5.

The simulation of spatial $\tau_s$ in CMIP6 is further evaluated against the empirical data with the additional statistical metrics. Modelled $\tau_s$ is found to be poorly spatially correlated to empirical $\tau_s$ in both the CMIP5 and CMIP6 models (shown by the curved axis in Fig. 4(c)). A slight increase in the ensemble mean spatial correlations is seen from CMIP5 (0.188) to CMIP6





(0.267), due to increases seen amongst individual models between CMIP5 and CMIP6, such as CESM2 from CCSM4, MPI-
ESM1.2-LR from MPI-ESM-LR, and NorESM2-LM from NorESM1-M. However, the consistency of modelled $\tau_s$ with the
benchmark datasets remains low. A particularly large range is seen in the spatial standard deviations of $\tau_s$ amongst the CMIP6
models, which is an increased range from CMIP5 (shown by the radial axis in Fig. 4(c)). The CMIP6 models with the most
extreme overestimations of the spatial standard deviations compared to the derived benchmark value (NorESM2-LM, CESM2,
and ACCESS-ESM1.5), are also found to have large RMSEs (Fig. 5(c)). Amongst the remaining CMIP6 models, the RMSEs
for modelled $\tau_s$ remain relatively consistent between CMIP5 and CMIP6.

### 3.4 Drivers of soil carbon spatial patterns: *Soil carbon spuriously highly correlated with NPP in CMIP5 and CMIP6*

#### 3.4.1 Global drivers

A negligible correlation ($\approx 0$) is found between the benchmark estimates of soil carbon and NPP, suggesting that soil carbon
is not spatially correlated to NPP in the real world. On the other hand, soil carbon and NPP ($C_s$-NPP) were found to be
significantly correlated in the models in both CMIP5 and CMIP6. The $C_s$-NPP spatial correlation was found to be greater
than 0.5 for six out of the ten CMIP5 ESMs and eight out of the eleven models in CMIP6 (Fig. 8(a)). However, a low spatial
correlation is found in the CMIP6 models CESM2 (0.134), NorESM2-LM (0.261), and BCC-CSM2-MR (0.214), values most
consistent with the benchmark datasets. The $C_s$-$\tau_s$ spatial correlations found in the CMIP6 models tend to underestimate the
positive correlation seen in the benchmark datasets (Fig. 8(a)). The majority of CMIP6 models see a negligible or slightly
negative $C_s$-$\tau_s$ spatial correlation, despite a low positive correlation produced by the benchmark datasets. The models BCC-
CSM2-MR, MIROC-ES2L, and NorESM2-LM are most consistent with the benchmark $C_s$-$\tau_s$ correlation.

   The modelled NPP to temperature (NPP-T) spatial correlations in CMIP6 are consistent with the positive relationship seen
in the benchmark datasets, however the magnitude of this positive correlation varies amongst the models (Fig. 8(b)). The
magnitude of the positive NPP-T correlation is underestimated in CanESM5, GFDL-ESM4, and NorESM2-LM, but otherwise
relatively consistent amongst the CMIP6 models. Nonetheless, a much greater range in the modelled NPP-T correlations
was seen amongst the CMIP5 models, suggesting an improved representation of this relationship in CMIP6. The variation in
modelled NPP-$\theta$ correlations remains in CMIP6, with models disagreeing in the sign and magnitude of the correlation of NPP
to soil moisture. The modelled NPP-$\theta$ correlation is the most consistent with the benchmark correlations in GFDL-ESM4,
MPI-ESM1.2-LR and UKESM1-0-LL (Fig. 8(b)).

It is generally agreed across the models in CMIP6 and CMIP5 that $\tau_s$ and temperature (T) are negatively correlated, with
the exception of MPI-ESM1.2-LR where a slight positive correlation is seen (Fig. 8(c)). This is consistent with the negative
$\tau_s$-T correlation derived in the benchmark dataset. There is variation amongst the models in the magnitude of the negative
correlation, with a significant overestimation seen in CanESM5. A negative correlation is also seen in the $\tau_s$-$\theta$ correlation
derived with the benchmark datasets. Inconsistencies with this empirical relationship are seen amongst the models in both
CMIP5 and CMIP6, with many negligible and positive correlations deduced (Fig. 8(c)). The exception is again MPI-ESM1.2-
LR, which in this case is the model most consistent with the benchmark $\tau_s$-$\theta$ correlation.





### 3.4.2 Regional drivers

The spatial correlations of modelled $C_s$-NPP are shown to be overestimated at every latitude in both CMIP6 and CMIP5, compared to the equivalent correlations derived from the empirical datasets. It can be seen that the CMIP6 ensemble mean
$C_s$-NPP correlation has an even larger positive bias compared to the benchmark correlation than in CMIP5. The empirical data sees a reduced $C_s$-NPP correlation in the northern latitudes, whereas a slight but less significant reduction is seen in the models (Fig. 9(a)). The spatial correlation between $C_s$-$\tau_s$ is shown to vary against latitude in the empirical datasets, where a greater correlation is seen in the tropical and northern latitude regions, and a negligible correlation is seen in the mid-latitudes (Fig. 9(b)). The CMIP6 models simulate the negligible $C_s$-$\tau_s$ seen in the mid-latitudes relatively consistently with the benchmark
data, where an improved consistency is seen from CMIP5. However, the CMIP6 models do not simulate the tropical and northern latitude positive $C_s$-$\tau_s$ correlations, where a negligible modelled correlation remains in these regions. CMIP5 is more consistent with the benchmark correlations than in CMIP6, where a positive modelled correlation $C_s$-$\tau_s$ is seen (Fig. 9(b)).

The spatial correlation between modelled soil carbon and soil moisture ($C_s$-$\theta$) is consistent with the correlations seen in the benchmark datasets at every latitude, with an improvement seen in the tropical correlation patterns in CMIP6 compared with
CMIP5 (Fig. 9(c)). Both the CMIP5 and CMIP6 ensembles span the benchmark $C_s$-$\theta$ correlation, though large model ranges in the $C_s$-$\theta$ sensitivity are seen across all latitudes. However, there is a reduced ensemble uncertainty in the $C_s$-$\theta$ correlation from CMIP5 to CMIP6 in low and mid latitudes. An overestimation of the negative $C_s$-T correlation seen in the benchmark datasets is present in both the CMIP5 and CMIP6 models, except the high latitudes (Fig. 9(d)). This modelled $C_s$-T correlation is particularly underestimated in the lower tropical latitudes, where a greater positive correlation is seen here in the benchmark
datasets. Fig. 9(d) suggests a slight improvement in the modelled tropical $C_s$-T correlation in CMIP6, and a worsening of modelled $C_s$-T in the high latitudes than in CMIP5, when compared to the $C_s$-T correlations in the benchmark datasets.

## 4 Discussion

### 4.1 Soil carbon stocks

#### 4.1.1 Global total soil carbon

Simulating global soil carbon stocks that are consistent with empirical data is required to predict reliable projections of future soil carbon storage and emission (Todd-Brown et al., 2013). Despite a reduced spread in model estimates of global total soil carbon within CMIP6 relative to CMIP5, discrepancies remain in the consistency of these estimates with the observations between the two CMIP generations. This together with the uncertainty associated with empirical datasets has resulted in no robust conclusion being drawn on the improvement of soil carbon simulation in CMIP6 compared to CMIP5. Due to
the potential significant feedback that exists between soil carbon and global climate, this lack of consistency may reduce our confidence in future projections of climate change (Friedlingstein et al., 2006; Gregory et al., 2009; Arora et al., 2013; Friedlingstein et al., 2014).



### 4.1.2 Spatial soil carbon patterns

Spatially, the simulation of soil carbon stocks sees some improvement between the CMIP5 and CMIP6 generations. Modelled
soil carbon was found to be poorly spatially correlated with the empirical data amongst models in both CMIP5 and CMIP6 (Fig.
4(a)). An improvement is seen on the spatial patterns across the mid-latitudes, which were generally overestimated in CMIP5.
However, significant underestimations of modelled soil carbon in the northern latitudes still remains which has a significant
impact on model predictions of global total soil carbon stocks (Fig. 1(a)). This systematic underestimation was previously
reported in the literature as a limitation of the CMIP5 models, where Todd-Brown et al. (2013) found northern latitude soil
carbon to be less consistent with the empirical data than on a global scale. This limitation remains amongst models in the
CMIP6 generation, where it was found that the majority of CMIP6 models underestimate northern latitude soil carbon stocks
regardless of whether or not the global soil carbon stocks are underestimated.

An exception to this northern latitude underestimation is seen within CMIP6 in the models CESM2 and NorESM2-LM.
These ESMs include the Land Surface Model (LSM) CLM5 (Lawrence et al., 2019), which is the first LSM to include the
representation of vertically resolved soil carbon in their CMIP simulations. This representation enables the inclusion of separate
carbon pools at varying depths in the soil, and allows for an improved simulation of soil carbon stocks (Koven et al., 2013). This
is of particular importance in the northern latitudes, where carbon stocks are expected to exist at much greater depths than the
1m considered in this study (Tarnocai et al., 2009; Ran et al., 2021). This can be seen in Table 3, where increased magnitudes
of soil carbon stocks are shown when increased depths are considered using the empirical datasets. A more thorough evaluation
of soil carbon in both CESM2 and NorESM2-LM is suggested for future research, with a particular focus on this improved
northern latitude soil carbon stocks simulation, however this evaluation of deeper soil carbon stocks (below 1m) is beyond the
scope of this study.

Accurately simulating soil carbon in the northern latitude regions is of particular importance as it is a major part of the total
global soil carbon pool (Jackson et al., 2017). Additionally, much of the carbon stored in these soil is held within permafrost,
which is known to be particularly sensitive to climate change. Permafrost thaw under climate change has the potential to release
significant amounts of carbon into the atmosphere over a short period of time with increased warming (Schuur et al., 2015;
Zimov et al., 2006; Burke et al., 2017; Hugelius et al., 2020), representing a significant feedback within the climate system.
Permafrost dynamics are generally poorly represented in ESMs, where Burke et al. (2020) found CMIP6 ESMs to have a similar
representation compared to CMIP5. Underestimating soil carbon in the northern latitudes may result in underestimating the
impact of this feedback in future climate change projections. Future improvements are needed to improve the simulation of
soil carbon stocks globally, but particularly within the northern latitudes.

### 4.2 Drivers of soil carbon change

To allow for a more in-depth understanding of the inconsistencies found between modelled and empirical soil carbon, the
simulation of above and below ground controls of soil carbon were also evaluated. Simulations of contemporary soil carbon
can be disaggregated into the effects of litter fall, which is well approximated by plant Net Primary Productivity (NPP), and




effective soil carbon turnover time ($\tau_s$), which is affected by both temperature and moisture of the soil (Koven et al., 2015). If models are to reliably simulate soil carbon in a way that is consistent with empirical data, the spatial drivers of soil carbon, NPP and $\tau_s$, must also be simulated consistently with empirical data. Isolating the effects of NPP and $\tau_s$ on soil carbon helps us to breakdown the simulation of soil carbon to help understand the limitations and inconsistencies seen amongst the models.

### 4.2.1 NPP

A robust improvement in the simulation of NPP is seen in the CMIP6 models compared with the CMIP5 models. This conclusion is deduced by: an increased number of models in CMIP6 have global total NPP values consistent with empirical data (Table 6), the overestimation of tropical NPP amongst CMIP5 models is seen to be reduced amongst the CMIP6 models (Fig. 1(b)), and more agreement is seen within CMIP6 relative to CMIP5 in the simulation of mid and northern latitude NPP (Fig. 3(b)). Modelled NPP was found to be robustly more consistent with the empirical data in CMIP6 compared with CMIP5 in all statistical evaluation metrics. Since CMIP5, multiple models have seen an addition of a dynamic nitrogen cycle (Davies-Barnard et al., 2020), where the models with nitrogen cycles are highlighted in Fig. 5 by the shaded bars. The results suggest an improvement in the simulation of NPP with the addition of dynamic nitrogen in models. However, CMIP6 models that do not represent a nitrogen cycle also mostly see improvements in the simulation of NPP, suggesting NPP is more constrained by observations in the most up to date generation of models. CanESM5 is the only model within CMIP6 to not see an overall improvement in the simulation of NPP, where NPP is found to be overestimated compared with the benchmark dataset. It is likely that the inclusion of a nitrogen cycle in this model would limit this overestimated NPP and improve consistency with the observations (Zhang et al., 2014; Exbrayat et al., 2013).

Despite an improved simulation of NPP in CMIP6, the spatial correlation between modelled soil carbon and NPP was found to be inconsistent with the equivalent empirically derived relationship. This result was previously shown for the CMIP5 models (Todd-Brown et al., 2013), and has been more recently shown for the CMIP6 models (Georgiou et al., 2021), both agreeing with the results found here. The majority of CMIP6 models were found to have positive $C_s$-NPP spatial correlations, opposed to a negligible spatial correlation found in the observations (Fig. 8(a)). Despite NPP driving the spatial pattern of soil carbon stocks, a positive correlation is not expected in the real world due to regions with high soil carbon not correlating with regions of high NPP. For example, in the observational derived data soil carbon stocks are greatest in the northern latitudes due to long turnover times in these regions, whereas NPP is lower due to cold temperatures in these regions limiting vegetation growth. The three CMIP6 models which did not significantly overestimate this correlation (CESM2, NorESM2-LM, and BCC-CSM2-MR), are three of the models with the most empirically consistent proportion of soil carbon stocks in the northern latitudes. Conversely the tropical regions see high NPP values, but warmer temperatures result in faster turnover times and lower soil carbon stocks. NPP is expected to increase in the future under climate change (Kimball et al., 1993; Friedlingstein et al., 1995; Amthor, 1995), which means an overly positive correlation in models could result in a subsequent increase in modelled projections of soil carbon stocks. An overestimation of future soil carbon storage could result in an overestimation of the future carbon sink and an inaccurate global carbon budget (Todd-Brown et al., 2013; Friedlingstein et al., 2020).



### 4.2.2 Soil carbon turnover time

The systematic improvements seen in the simulation of NPP in CMIP6 are not seen in the simulation of soil carbon turnover time ($\tau_s$), where the simulation of $\tau_s$ is found to remain inconsistent with the empirical data in CMIP6. Improvements are seen within CMIP6 relative to CMIP5, such as more agreement within the ensemble in the mid-latitudes and tropical regions, however less agreement is seen in the northern latitudes (Fig. 3(c)). Northern latitude $\tau_s$ is generally underestimated in models, which corresponds to the underestimation of soil carbon seen in these regions. This has been previously identified in ESMs,

where it was found that the underestimation of global $\tau_s$ amongst the CMIP5 models is primarily due to low values in the northern latitudes (Wu et al., 2018). The reduced agreement in CMIP6 is due to long $\tau_s$ values existing in the northern latitudes of CESM2 and NorESM2-LM, alongside the general ensemble underestimations (Fig. 7). The increased northern latitude $\tau_s$ values in CESM2 and NorESM2-LM is likely to be due to the improved representation of soil carbon pools, where vertically resolved soil carbon allows for differential $\tau_s$ values for pools at varying depths. Despite these individual improvements since

CMIP5, large discrepancies exist within the CMIP6 ensemble between modelled and empirical $\tau_s$.

To simulate $\tau_s$ consistently with observations, the relationship of $\tau_s$ to both temperature (T) and moisture ($\theta$) must also be simulated in a way that is consistent with observations. Generally, the $\tau_s$-T relationship is consistently simulated, however there is variation in the modelled temperature sensitivity of $\tau_s$ across the ensemble. The $\tau_s$-$\theta$ relationship is less consistently represented, where the majority of CMIP6 models do not match the empirically derived relationship. Despite a positive dependence

of soil respiration on soil moisture, many of the CMIP6 models display a contradictory positive $\tau_s$-$\theta$ correlation (Fig. 8). This lack of consistency between the modelled and empirical relationships involving $\tau_s$, is likely to be due to key soil processes not being represented. Particularly, a limitation of the $\tau_s$-$\theta$ relationship in ESMs is the representation of peat not being simulated. Peat forms in wet areas globally, so to simulate the $\tau_s$ relationship to soil moisture consistently with empirical data, models must simulate increased, longer turnover times in regions where peat exists. Moreover, to accurately simulate the accumulation

of peat in models, the soil column must be vertically resolved to allow for the soil column to grow (Chadburn et al., 2021).

These results suggest much of the uncertainty associated with modelled soil carbon stocks can be attributed to the simulation of below ground processes. The improved consistency of NPP with empirical data suggests considerable efforts have been made to achieve an improved representation of above ground processes in CMIP6 ESMs since the release of the CMIP5 ensemble. However, the same improvements are not seen in the simulation of $\tau_s$ as systemic limitations remain in the new generation of

models, suggesting the same progress on the model development of below ground processes has not been achieved between CMIP5 and CMIP6. Moreover, focus on above ground processes without consideration of below ground processes can result in inconsistencies of soil carbon stocks. For example, the inclusion of a nitrogen cycle has been shown to lead to a reduction in soil carbon in the model, see Fig. 6 in Wiltshire et al. (2021), so tuning of the baseline turnover rates is required to keep soil carbon stocks consistent with observed values.

The required improvement of soil carbon pool turnover rates has previously been identified for the CMIP5 ensemble (Nishina et al., 2014), and more recently, Ito et al. (2020) find that the difference in turnover times amongst the CMIP6 models is responsible for approximately 88% of the variation seen in global soil carbon stocks amongst the models, and state that





constraining key parameters which control soil carbon turnover processes is a key area for future model development. A key development seen in CMIP6 since CMIP5 is the representation of vertically resolved soil carbon. Models which simulate non-
vertically resolved soil carbon typically turn over all the carbon based on the temperature near the soil surface. This could lead to reduced quantities of soil carbon and an underestimation of northern latitude soil carbon stocks, due to near surface soil being warmer than the deeper soil, and as turnover is known to respond exponentially to temperature (Davidson and Janssens, 2006). Overall, further improvements in the representation of soil carbon turnover time, with a particular focus on the northern latitudes, is identified as a key area for future model development.

## 5   Conclusions

The ability of Earth System Models (ESMs) to simulate present day soil carbon is vital to help produce reliable projections of climate change. In this study the most up-to-date ESMs, which are part of the CMIP6 ensemble, have been evaluated against empirical datasets to assess their ability to represent soil carbon and related controls: Net Primary Productivity (NPP) and the effective soil carbon turnover time ($\tau_s = C_s/R_h$). The evaluation is completed by comparison to the previous generation of
CMIP5 ESMs, to assess where improvements have been made and to identify priorities for future model development. Studies of this type rely on the provided CMIP and empirical data, for which we are thankful. Below the key conclusions from this study are listed:

1. The spatial patterns of soil carbon in CMIP6 models are more in agreement with each other than they were in CMIP5, and are more consistent with observations in the mid-latitudes. However, soil carbon is still heavily underestimated in
high northern latitudes (with the exception of two CMIP6 models that represent deep soil carbon).

2. Overall no significant improvements are seen in the simulation of the observed spatial pattern of soil carbon across the globe from the CMIP5 to the CMIP6 generation.

3. There is good evidence that spatial patterns of contemporary NPP are better simulated in CMIP6 than in CMIP5 generation models, when compared to satellite-derived estimates.

4. However, spatial patterns of $\tau_s$ continue to be poorly represented in CMIP6 models, in comparison to estimates derived from observational datasets of soil carbon and heterotrophic respiration ($R_h$).

5. Importantly, soil carbon simulations in both the CMIP5 and CMIP6 ESM generations seem to be spuriously highly-correlated with NPP, which may make soil carbon in these models over responsive to future projected changes in NPP.

Taken together, these conclusions point to a need for a much greater emphasis on improving the representation of below-ground
soil processes in next generation (CMIP7) of ESMs.

*Data availability.* The datasets analysed during this study are available online: CMIP5 model output [https://esgf-node.llnl.gov/search/cmip5/], CMIP6 model output [https://esgf-node.llnl.gov/search/cmip6/], Harmonized World Soils Database (HWSD) and Northern Circumpolar Soil Carbon Database (NCSCD) [https://github.com/rebeccamayvarney/CMIP_soilcarbon_evaluation], World Inventory of Soil property Estimates (WISE30sec) [https://www.isric.org/explore/wise-databases], Sanderman et al.
2017 soil carbon estimate (1m and 2m) [https://dataverse.harvard.edu/dataset.xhtml?persistentId=doi:10.7910/DVN/QQQM8V], Global Soil Dataset for use in Earth System Models (GSDE) [http://globalchange.bnu.edu.cn/research/soilw], Global Gridded Surfaces of Selected Soil Characteristics (IGBP-DIS) [https://daac.ornl.gov/cgi-bin/dsviewer.pl?ds_id=569], MODIS Net Primary Production [https://lpdaac.usgs.gov/products/mod17a3v055], CARDAMOM Heterotrophic Respiration [https://datashare.is.ed.ac.uk/handle/10283/875], Copernicus Climate Change Service (C3S) soil
moisture gridded dataset [https://cds.climate.copernicus.eu/cdsapp#!/dataset/eu.copernicus.climate.satellite-soil-moisture?tab=overview&utm_medium=chatbot&utm_source=cds], and the WFDEI Meteorological Forcing Data [https://rda.ucar.edu/datasets/ds314.2/].

*Author contributions.* R.M.V., S.E.C., and P.M.C. outlined the evaluation and drafted the manuscript, and R.M.V. completed the analysis and produced the figures. E.J.B. provided the empirical datasets and gave helpful advice to address the empirical uncertainty. All co-authors
provided guidance on the study at various times and suggested edits to the draft manuscript.

*Competing interests.* The authors declare that they have no competing interests.

*Acknowledgements.* This work was supported by the European Research Council 'Emergent Constraints on Climate-Land feedbacks in the Earth System (ECCLES)' project, grant agreement number 742472 (R.M.V. and P.M.C.). S.E.C. was supported by a Natural Environment Research Council independent research fellowship, grant no. NE/R015791/1. E.J.B. was supported by the Joint UK BEIS/Defra Met Office
Hadley Centre Climate Programme (grant no. GA01101). We acknowledge the World Climate Research Programme's Working Group on Coupled Modelling, which is responsible for CMIP, and we thank the climate modelling groups for producing and making their model output available to enable studies such as this evaluation study. We also thank providers of empirical datasets, which enabled us to complete the evaluation in this study. The Taylor diagrams presented in this study was produced using code at [https://gist.github.com/ycopin/3342888].



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



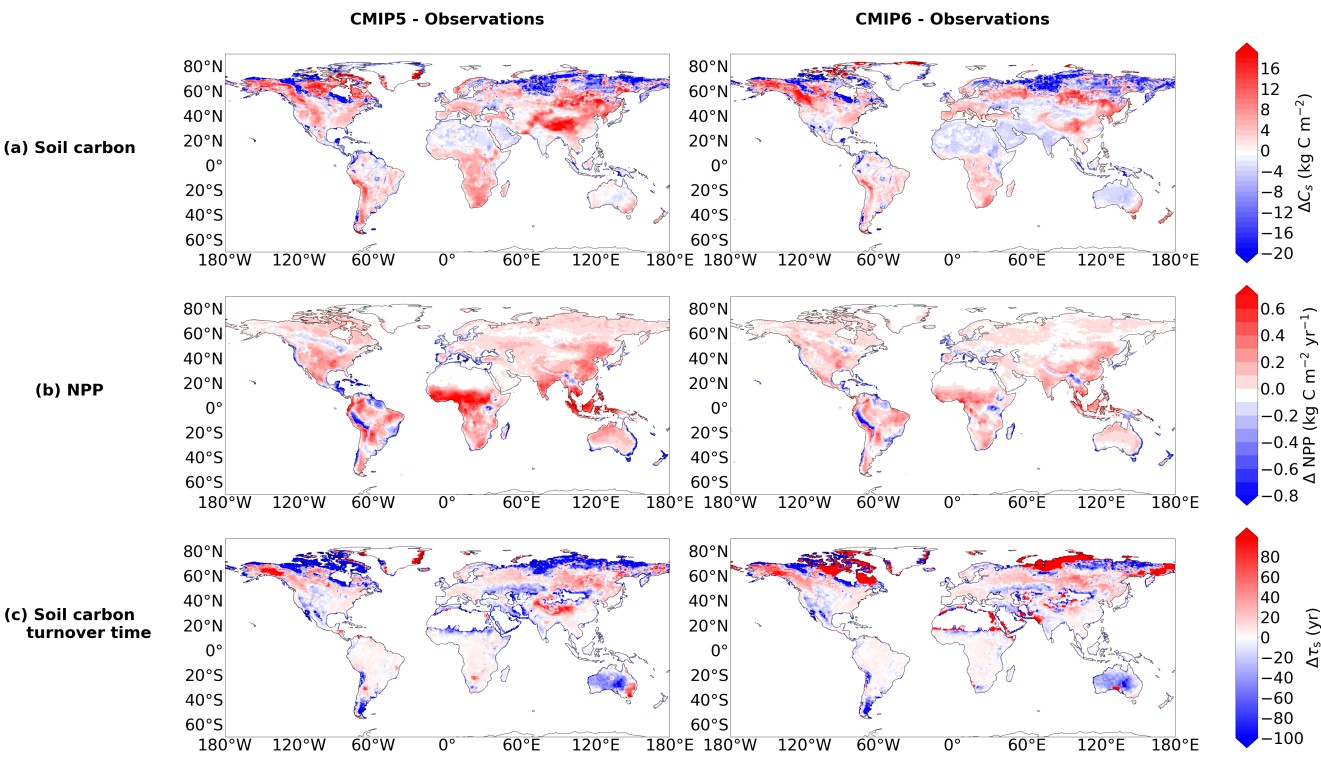

**Figure 1.** Maps presenting the difference between the modelled and benchmark data for the CMIP5 and CMIP6 ensembles, for: (a) $C_s$ (kg m$^{-2}$), (b) NPP (kg m$^{-2}$ yr$^{-1}$), and (c) $\tau_s$ (yr).



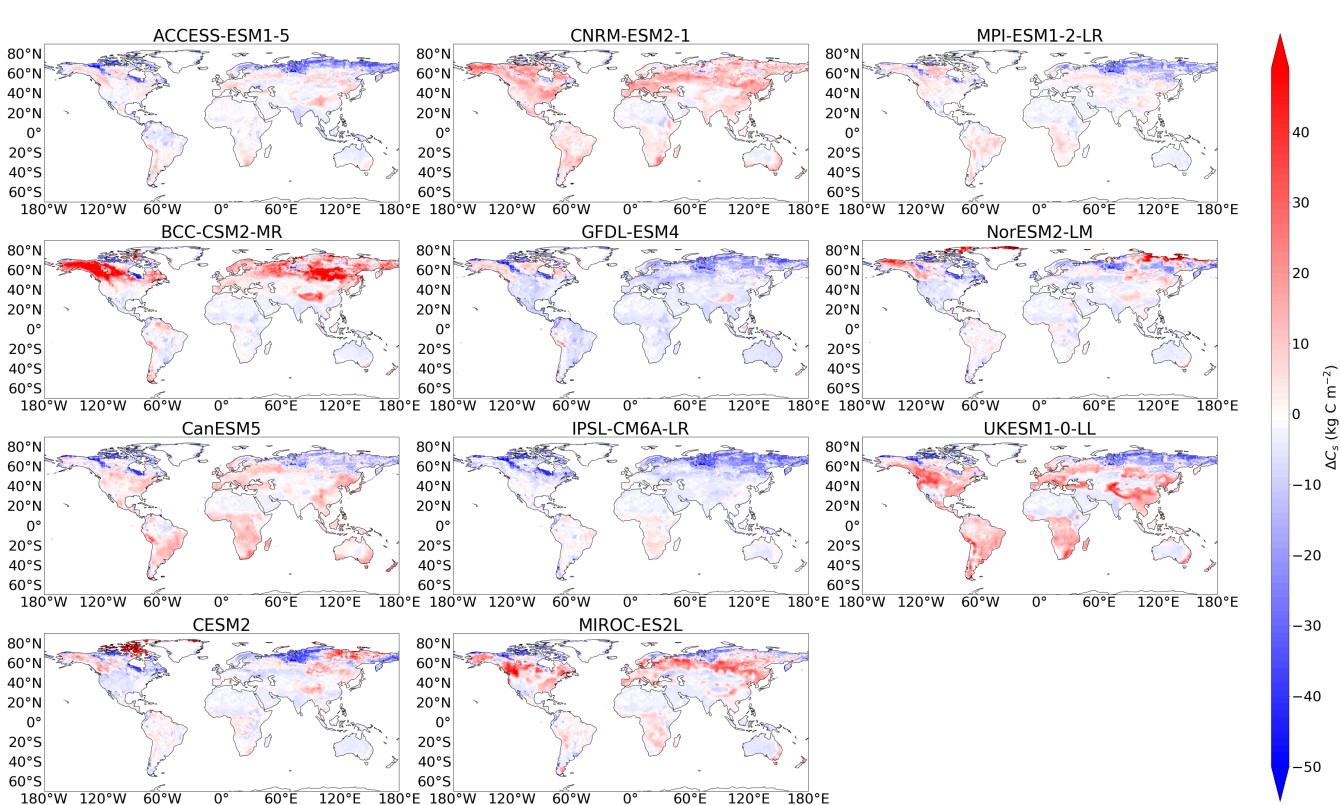

**Figure 2.** Maps of the difference in soil carbon ($C_s$) between the historical simulation of each CMIP6 model and the benchmark data.





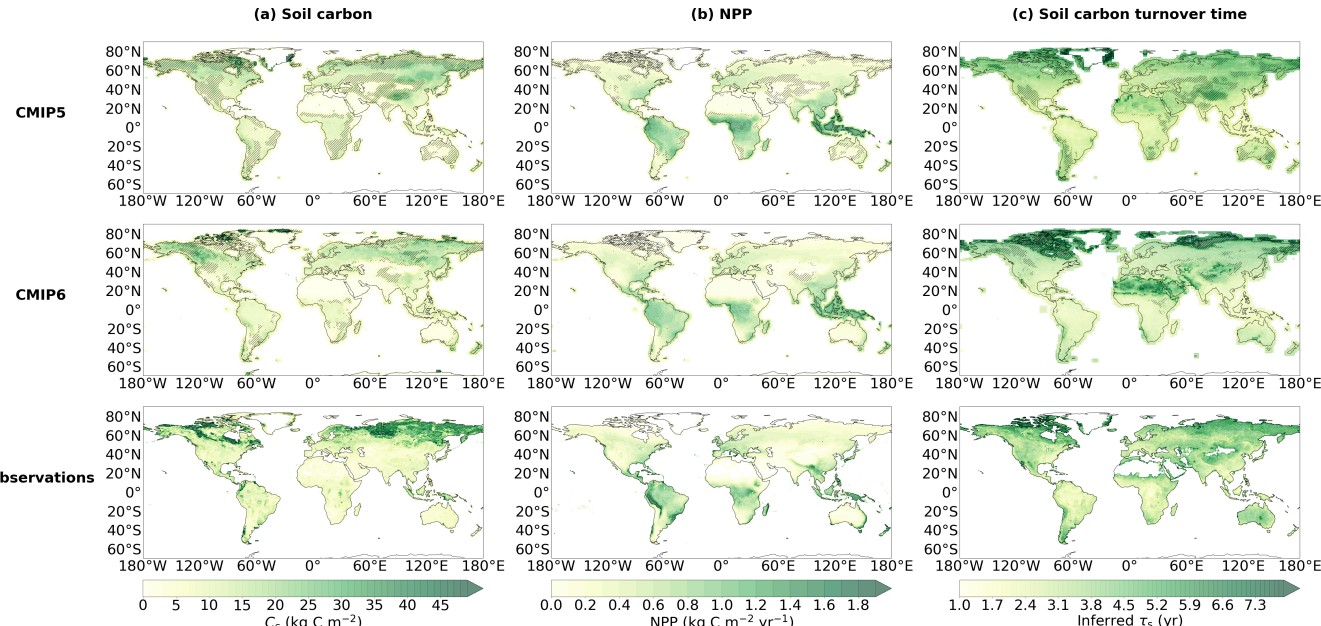

**Figure 3.** Ensemble mean maps for (a) $C_s$ (kg m$^{-2}$), (b) NPP (kg m$^{-2}$ yr$^{-1}$), and (c) $\tau_s$ (yr), presented for the CMIP6 ensemble, CMIP5 ensemble and the benchmark datasets. The hatched areas are used to show regions of low agreement within the ensemble, where regions of low soil carbon ($< 5$ kg m$^{-2}$) have been excluded.


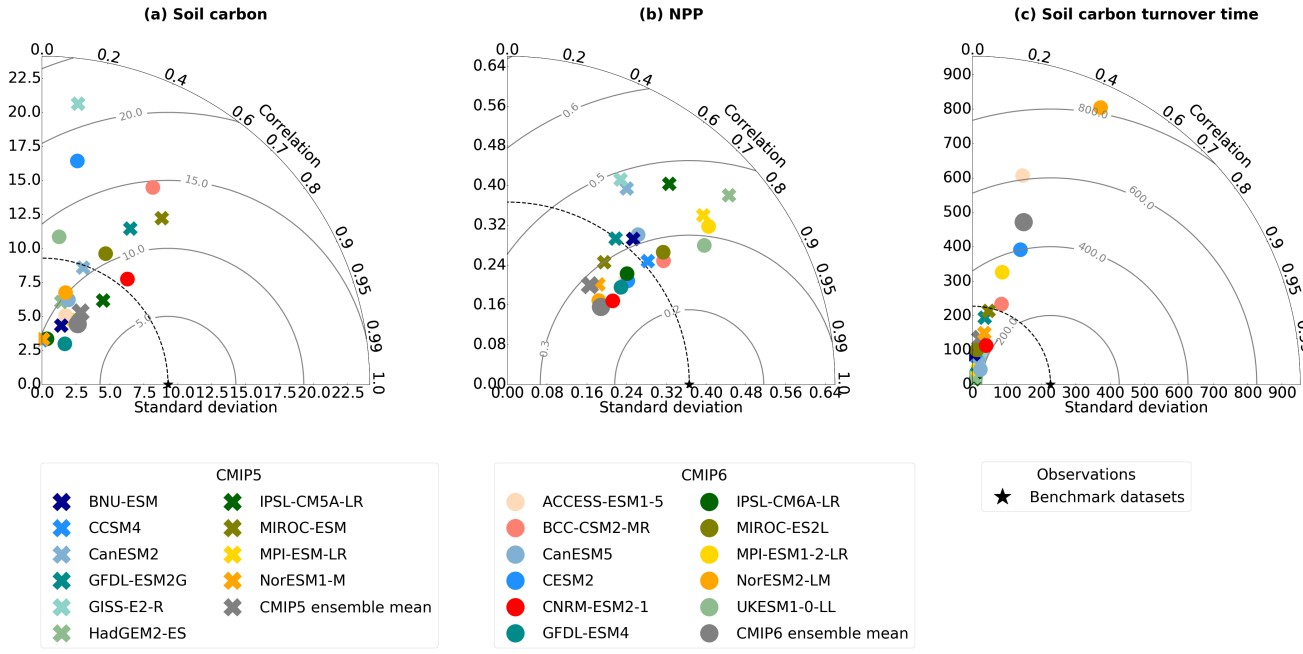

**Figure 4.** Taylor diagrams showing the spatial standard deviation (shown by the radial axis between standard x and y axes), the Pearson correlation coefficients (shown by the curved correlation axis), and the RMSE (show by the grey contours), for the ESMs in both CMIP5 and CMIP6 compared to the benchmark datasets, for (a) soil carbon ($C_s$), (b) NPP, and (c) soil carbon turnover time ($\tau_s$).



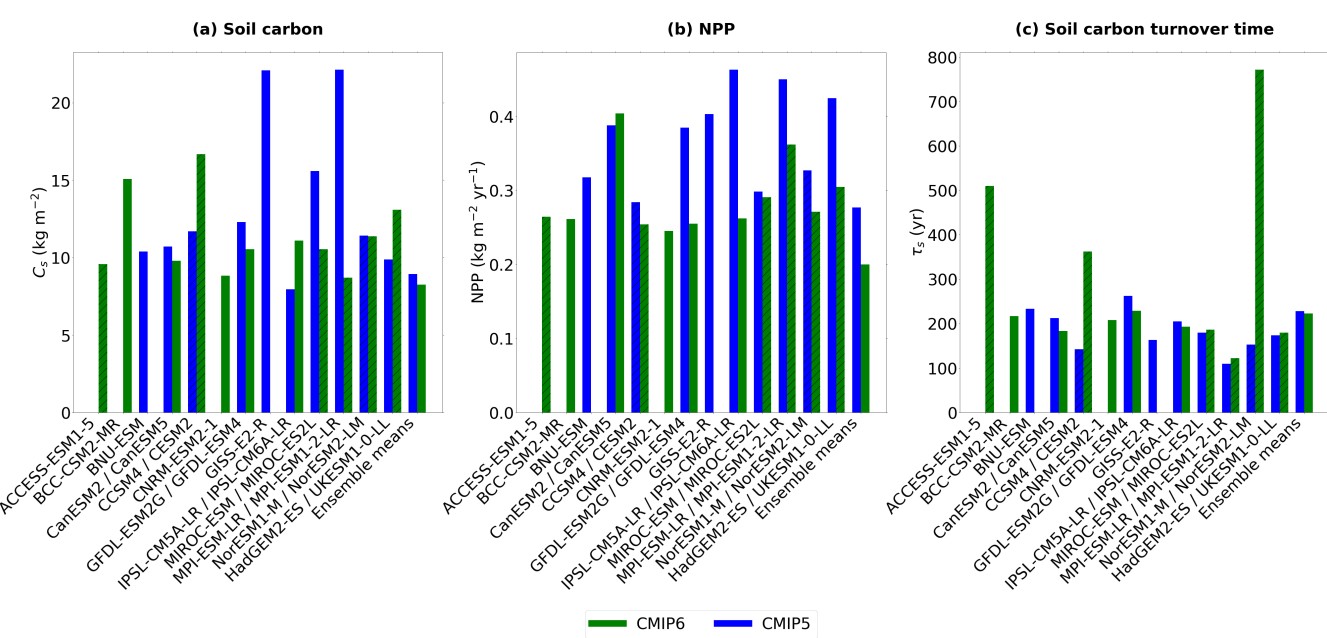

**Figure 5.** Bar charts comparing the Root Mean Squared Errors (RMSEs) in CMIP6 and CMIP5, for (a) soil carbon ($C_s$), (b) NPP, and (c) soil carbon turnover time ($\tau_s$).



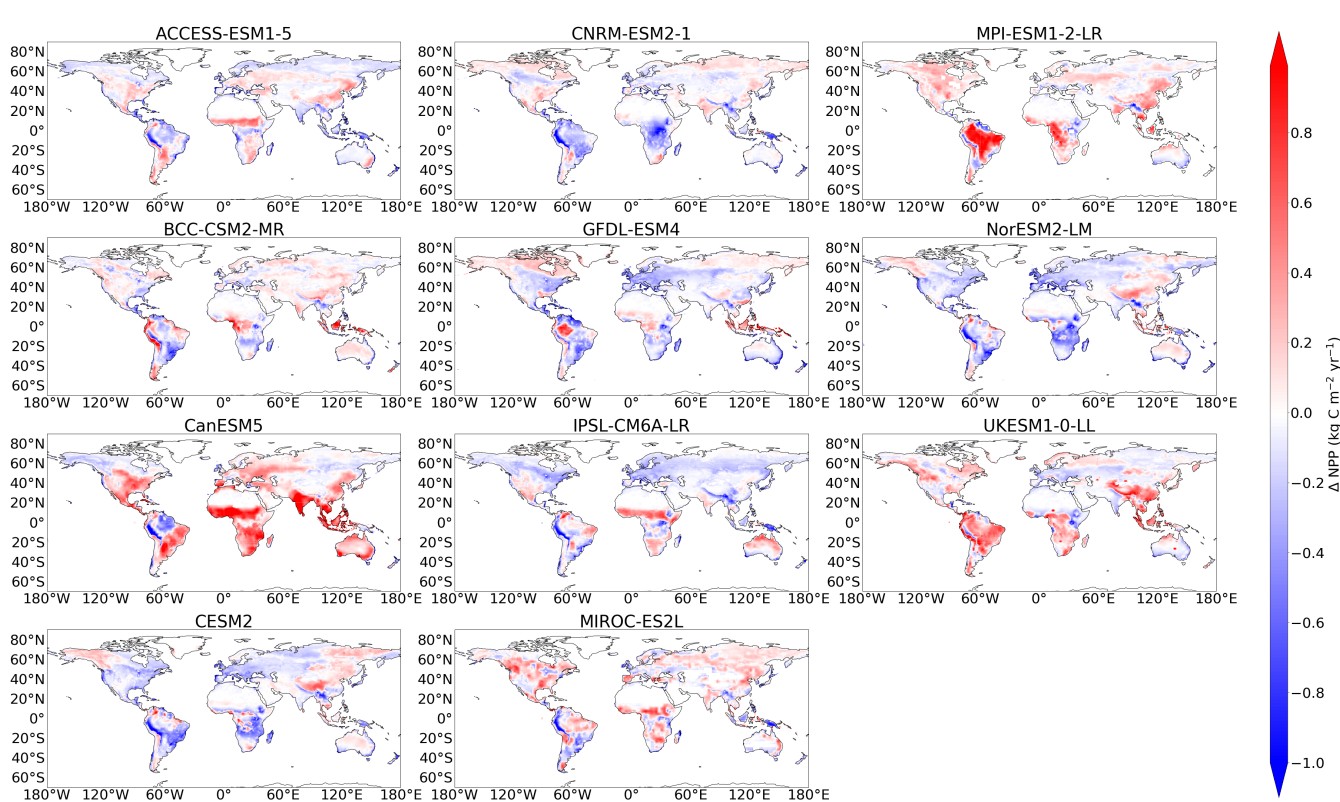

**Figure 6.** Maps of the difference in Net Primary Production (NPP) between the historical simulation of each CMIP6 model and the benchmark dataset.





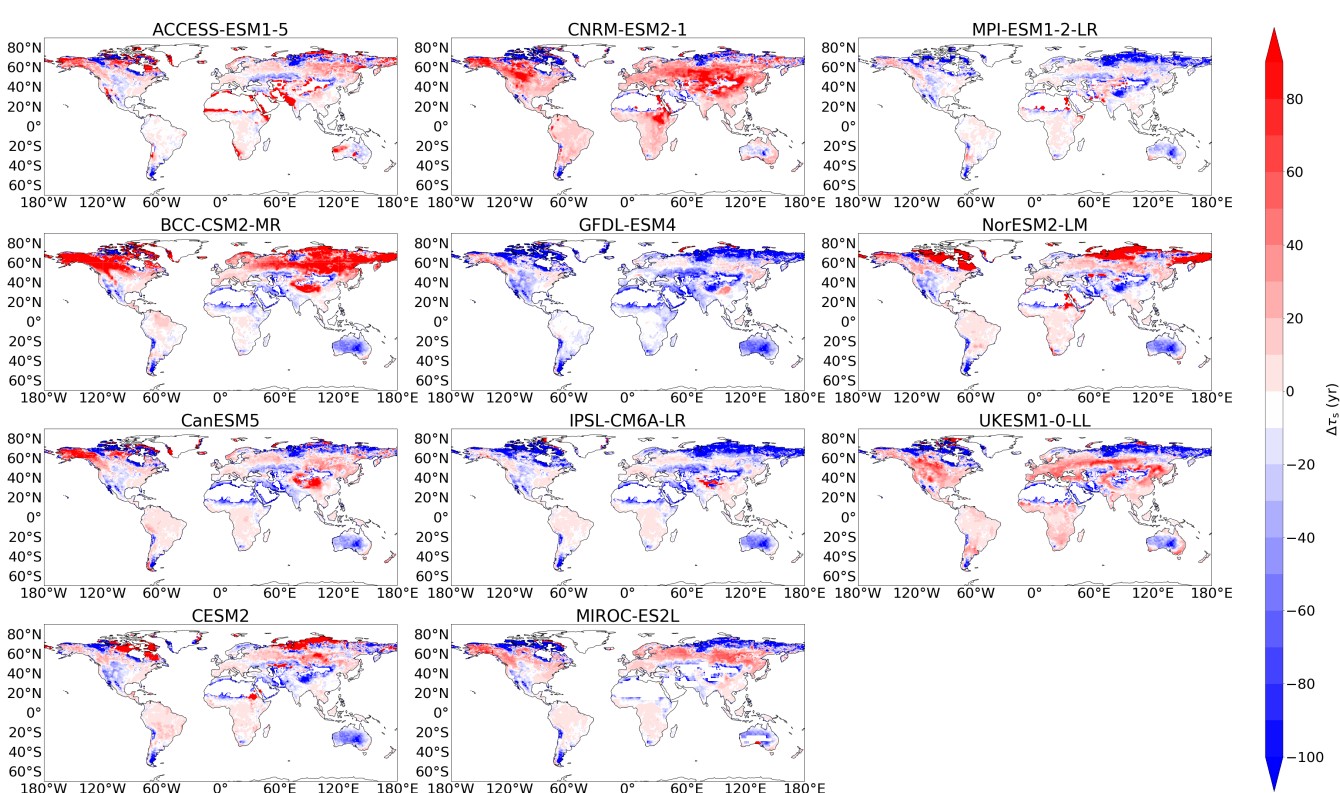

**Figure 7.** Maps of the difference in soil carbon turnover time ($\tau_s$) between the historical simulation of each CMIP6 model and the benchmark datasets.





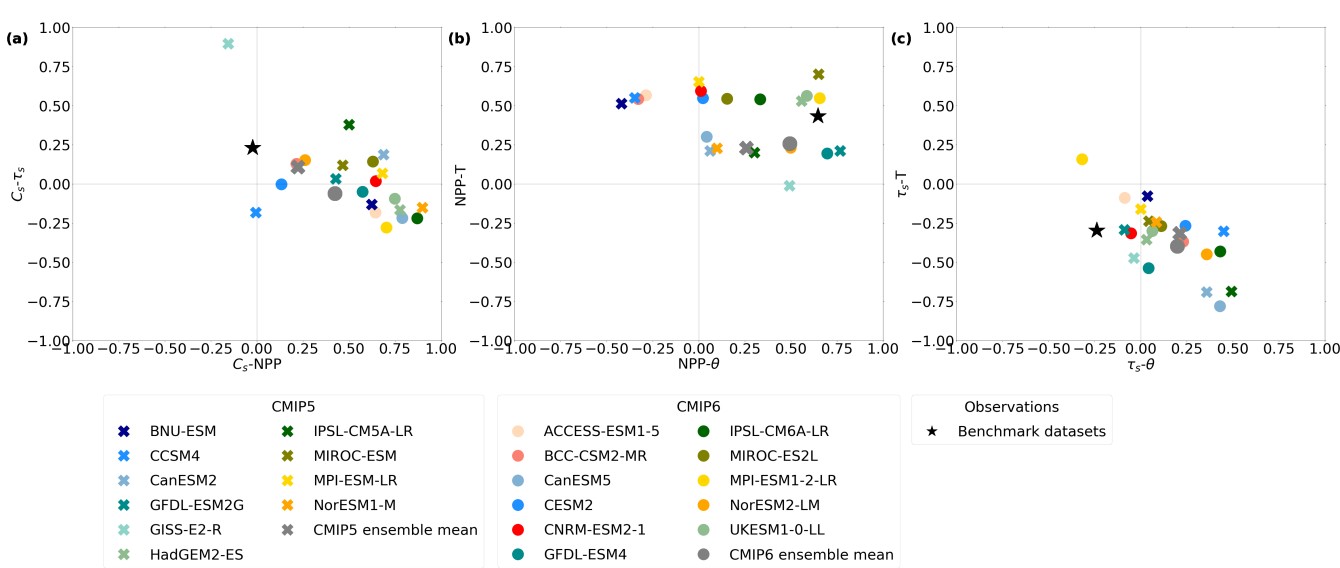

**Figure 8.** Scatter plots investigating the relationships between different Pearson correlation coefficients of climate variables, (a) $C_s$-$\tau_s$ against $C_s$-NPP, (b) NPP-T against NPP-$\theta$, (c) $\tau_s$-T against $\tau_s$-$\theta$.





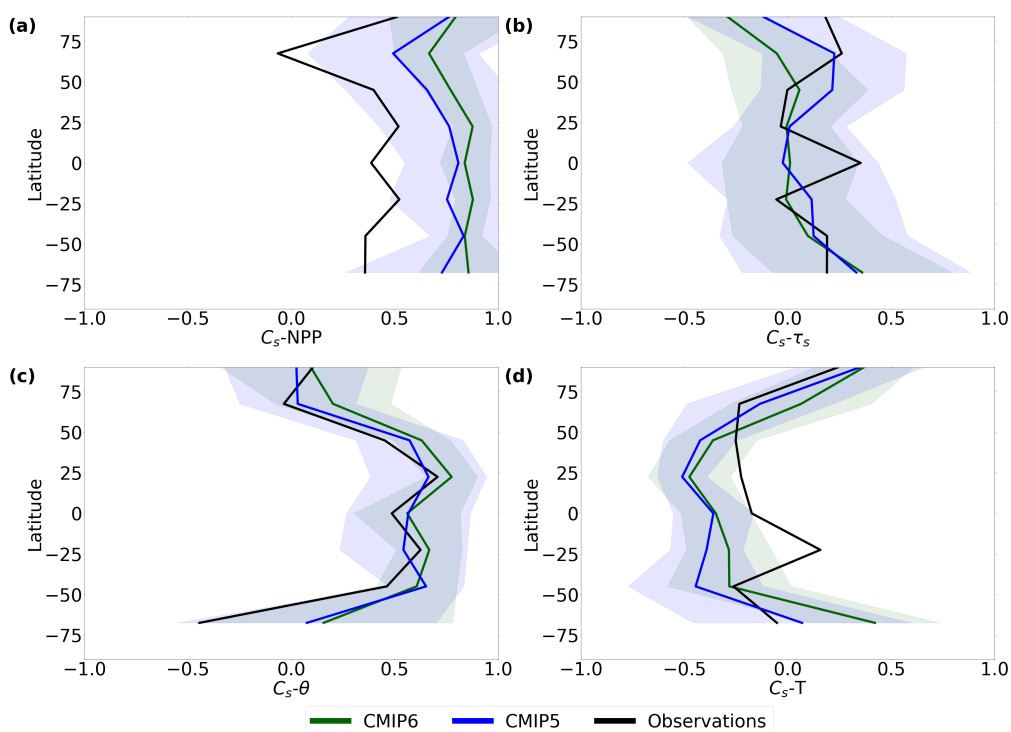

**Figure 9.** The latitudinal profiles of the Pearson correlation coefficients between soil carbon and (a) NPP ($C_s$-NPP), (b) soil carbon turnover time ($C_s$-$\tau_s$), (c) soil moisture ($C_s$-$\theta$), and (d) temperature ($C_s$-T).

**Table 1.** The 11 CMIP6 Earth System Models included in this study, and relevant features of their land carbon cycle components (Arora et al., 2020).

| Earth System Model | Modelling Centre | Land Surface Model | Nitrogen cycle | No. of live carbon pools | No. of dead carbon pools | Temperature & Moisture | References |
|---|---|---|---|---|---|---|---|
| ACCESS-ESM1.5 | CSIRO | CABLE2.4 + CASA-CNP | Yes | 3 | 6 | Arrhenius & Hill | Ziehn et al. (2020) Haverd et al. (2018) Trudinger et al. (2016) |
| BCC-CSM2-MR | BCC | BCC-AVIM2 | No | 3 | 8 | Hill & Hill | Wu et al. (2019) Ji et al. (2008) |
| CanESM5 | CCCma | CLASS-CTEM | No | 3 | 2 | $Q_{10}$ & Hill | Swart et al. (2019) Melton et al. (2020) Seiler et al. (2021) |
| CESM2 | CESM | CLM5 | Yes | 22 | 7 | Arrhenius & Increasing | Danabasoglu et al. (2020) Lawrence et al. (2019) |
| CNRM-ESM2-1 | CNRM | ISBA-CTRIP | No | 6 | 7 | $Q_{10}$ & Increasing | Séférian et al. (2019) Delire et al. (2020) |
| GFDL-ESM4 | GFDL | LM4.1 | No | 6 | 4 | Hill & Increasing | Dunne et al. (2020) Zhao et al. (2018) |
| IPSL-CM6A-LR | IPSL | ORCHIDEE branch 2.0 | No | 8 | 3 | $Q_{10}$ & Increasing | Boucher et al. (2020) Cheruy et al. (2020) Guimberteau et al. (2018) |
| MIROC-ES2L | JAMSTEC | MATSIRO VISIT-s | Yes | 3 | 6 | Arrhenius & Increasing | Hajima et al. (2020) Ito and Oikawa (2002) |
| MPI-ESM1.2-LR | MPI | JSBACH3.2 | Yes | 3 | 18 | $Q_{10}$ & Increasing | Mauritsen et al. (2019) Goll et al. (2017) Goll et al. (2015) |
| NorESM2-LM | NCC | CLM5 | Yes | 22 | 7 | Arrhenius & Increasing | Seland et al. (2020) Lawrence et al. (2019) |
| UKESM1-0-LL | UK | JULES-ES-1.0 | Yes | 3 | 4 | $Q_{10}$ & Hill | Sellar et al. (2020) Wiltshire et al. (2021) |



**Table 2.** The 10 CMIP5 Earth System Models included in this study, and relevant features of their land carbon cycle components (Arora et al., 2013; Anav et al., 2013; Friedlingstein et al., 2014). Including temperature and moisture functions presented in Todd-Brown et al. (2013).

| Earth System Model | Modelling Centre | Land Surface Model | Nitrogen cycle | No. of live & dead carbon pools | Temperature & Moisture | References |
|---|---|---|---|---|---|---|
| BNU-ESM | BNU | CoLM + BNU-DGVM | Yes | - | $Q_{10}$ & Increasing | Ji et al. (2014) Dai et al. (2003) |
| CCSM4 | CCSM | CLM4 | Yes | 20 | Arrhenius & Increasing | Gent et al. (2011) Lawrence et al. (2011) |
| CanESM2 | CCCma | CLASS2.7 + CTEM1 | No | 5 | $Q_{10}$ & Hill | Arora et al. (2009) Arora and Boer (2010) |
| GFDL-ESM2G | GFDL | LM3 | No | 10 | Hill & Increasing | Dunne et al. (2012) Dunne et al. (2013) Shevliakova et al. (2009) |
| GISS-E2-R | NASA-GISS | YIBs, version 1.0 | No | 12 | Increasing & Increasing | Schmidt et al. (2014) Yue and Unger (2015) |
| HadGEM2-ES | MOHC | JULES + TRIFFID | No | 7 | $Q_{10}$ & Hill | Jones et al. (2011) Best et al. (2011) Clark et al. (2011) |
| IPSL-CM5A-LR | IPSL | ORCHIDEE | No | 7 | $Q_{10}$ & Increasing | Dufresne et al. (2013) Krinner et al. (2005) |
| MIROC-ESM | JAMSTEC | MATSIRO + SEIB-DGVM | No | 6 | Arrhenius & Increasing | Watanabe et al. (2011) Ito and Oikawa (2002) Sato et al. (2007) |
| MPI-ESM-LR | MPI | JSBACH + BETHY | No | 6 | $Q_{10}$ & Increasing | Raddatz et al. (2007) Knorr (2000) |
| NorESM1-M | NCC | CLM4 | Yes | 20 | Arrhenius & Increasing | Bentsen et al. (2013) Iversen et al. (2013) Lawrence et al. (2011) |





**Table 3.** Table of global total and northern latitude total (northern latitudes defined as 60° N - 90° N) soil carbon estimates from multiple empirical datasets, for varying soil depths where applicable.

| Empirical dataset | Depth | Global total $C_s$ (PgC) | Northern latitude total $C_s$ (PgC) | Reference |
|---|---|---|---|---|
| HWSD + NCSCD | 1m | 1412 ± 215 | 401 ± 61 | FAO and ISRIC (2012) Hugelius et al. (2013) |
| WISE30sec | 1m | 1371 ± 129 | 314 | Batjes (2016) |
|  | 2m | 1952 ± 198 | 468 |  |
| S2017 | 1m | 1966 | 515 | Sanderman et al. (2017) |
|  | 2m | 3141 | 893 |  |
| GSDE | 1m | 1682 | 526 | Shangguan et al. (2014) |
|  | 2.3m | 2593 | 849 |  |
| IGBP DIS | 1m | 1567 | 377 | IGBP (2000) |



**Table 4.** Table presenting global soil carbon values for the 11 CMIP6 models included in this study and the benchmark datasets. Including: global total $C_s$ in PgC, and northern latitude total (90°N - 60°N) $C_s$ in PgC, and the spatial mean value of $C_s$ and corresponding standard deviation in kg m$^{-2}$.

| Earth System Model | Global total $C_s$ (PgC) | Northern latitude total $C_s$ (PgC) | Mean $C_s$ ± std (kg m$^{-2}$) |
|---|---|---|---|
| ACCESS-ESM1.5 | 900 | 151 | 5.86 ± 5.35 |
| BCC-CSM2-MR | 1770 | 575 | 11.6 ± 16.6 |
| CanESM5 | 1500 | 218 | 3.87 ± 6.52 |
| CESM2 (cSoilAbove1m) | 991 | 294 | 7.05 ± 16.6 |
| *CESM2 (cSoil)* | *1870* | *1036* | *13.8 ± 51.7* |
| CNRM-ESM2-1 | 1810 | 440 | 12.2 ± 9.98 |
| GFDL-ESM4 | 516 | 163 | 1.36 ± 3.43 |
| IPSL-CM6A-LR | 639 | 66.0 | 4.80 ± 3.37 |
| MIROC-ES2L | 1460 | 347 | 9.31 ± 10.7 |
| MPI-ESM1.2-LR | 970 | 175 | 6.68 ± 5.23 |
| NorESM2-LM (cSoilAbove1m) | 969 | 300 | 2.61 ± 6.97 |
| *NorESM2-LM (cSoil)* | *2430* | *1563* | *6.60 ± 41.3* |
| UKESM1-0-LL | 1760 | 194 | 12.0 ± 10.9 |
| Ensemble mean | 1206 ± 445 | 266 ± 139 | 2.80 ± 5.15 |
| Benchmark dataset | 1412 ± 215 | 401 ± 83 | 10.7 ± 9.28 |



**Table 5.** Table presenting global soil carbon values for the 10 CMIP5 models included in this study and the benchmark datasets. Including: global total $C_s$ in PgC, and northern latitude total (90°N - 60°N) $C_s$ in PgC, and the spatial mean value of $C_s$ and corresponding standard deviation in kg m$^{-2}$.

| Earth System Model | Global total $C_s$ (PgC) | Northern latitude total $C_s$ (PgC) | Mean $C_s \pm$ std (kg m$^{-2}$) |
|---|---|---|---|
| BNU-ESM | 681 | 135 | $5.31 \pm 4.55$ |
| CCSM4 | 507 | 28.1 | $4.03 \pm 3.24$ |
| CanESM2 | 1540 | 300 | $9.16 \pm 9.11$ |
| GFDL-ESM2G | 1420 | 635 | $9.47 \pm 13.2$ |
| GISS-E2-R | 2150 | 609 | $15.9 \pm 20.8$ |
| HadGEM2-ES | 1080 | 148 | $8.19 \pm 6.24$ |
| IPSL-CM5A-LR | 1350 | 346 | $9.77 \pm 7.64$ |
| MIROC-ESM | 2550 | 742 | $20.5 \pm 15.1$ |
| MPI-ESM-LR | 3000 | 204 | $23.5 \pm 14.8$ |
| NorESM1-M | 538 | 31.0 | $3.61 \pm 3.34$ |
| Ensemble mean | $1480 \pm 810$ | $318 \pm 246$ | $10.5 \pm 6.02$ |
| Benchmark dataset | $1412 \pm 215$ | $401 \pm 83$ | $10.7 \pm 9.28$ |

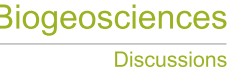


**Table 6.** Table presenting global carbon fluxes and turnover time values for the 11 CMIP6 models included in this study and the benchmark datasets. Including: global total NPP (PgC yr$^{-1}$) and effective average soil carbon turnover time (yr).

| Earth System Model | NPP (PgC yr$^{-1}$) | $\tau_s$ (yr) |
|---|---|---|
| ACCESS-ESM1.5 | 45.6 | 19.0 |
| BCC-CSM2-MR | 51.2 | 34.1 |
| CanESM5 | 75.5 | 18.1 |
| CESM2 (cSoilAbove1m) | 43.9 | 25.8 |
| *CESM2 (cSoil)* | - | 50.4 |
| CNRM-ESM2-1 | 45.6 | 41.5 |
| GFDL-ESM4 | 52.6 | 11.2 |
| IPSL-CM6A-LR | 46.4 | 14.6 |
| MIROC-ES2L | 59.1 | 24.5 |
| MPI-ESM1.2-LR | 58.9 | 15.4 |
| NorESM2-LM (cSoilAbove1m) | 43.5 | 24.0 |
| *NorESM2-LM (cSoil)* | - | *60.8* |
| UKESM1-0-LL | 60.8 | 28.1 |
| Ensemble mean | 53.0 ± 9.39 | 23.3 ± 8.59 |
| Benchmark datasets | 56.6 ± 14.3 | $27.0^{+27}_{-11}$ |





**Table 7.** Table presenting global carbon fluxes and turnover time values for the 10 CMIP5 models included in this study and the benchmark datasets. Including: global total NPP (PgC yr$^{-1}$) and effective average soil carbon turnover time (yr).

| Earth System Model | NPP (PgC yr$^{-1}$) | $\tau_s$ (yr) |
|---|---|---|
| BNU-ESM | 44.3 | 16.6 |
| CCSM4 | 42.9 | 14.3 |
| CanESM2 | 59.0 | 72.9 |
| GFDL-ESM2G | 74.4 | 57.3 |
| GISS-E2-R | 31.0 | 47.1 |
| HadGEM2-ES | 69.1 | 16.8 |
| IPSL-CM5A-LR | 76.6 | 19.4 |
| MIROC-ESM | 47.1 | 56.8 |
| MPI-ESM-LR | 73.5 | 42.7 |
| NorESM1-M | 45.0 | 34.5 |
| Ensemble mean | 56.3 ± 15.4 | 37.8 ± 19.7 |
| Benchmark datasets | 56.6 ± 14.3 | $27.0^{+27}_{-11}$ |





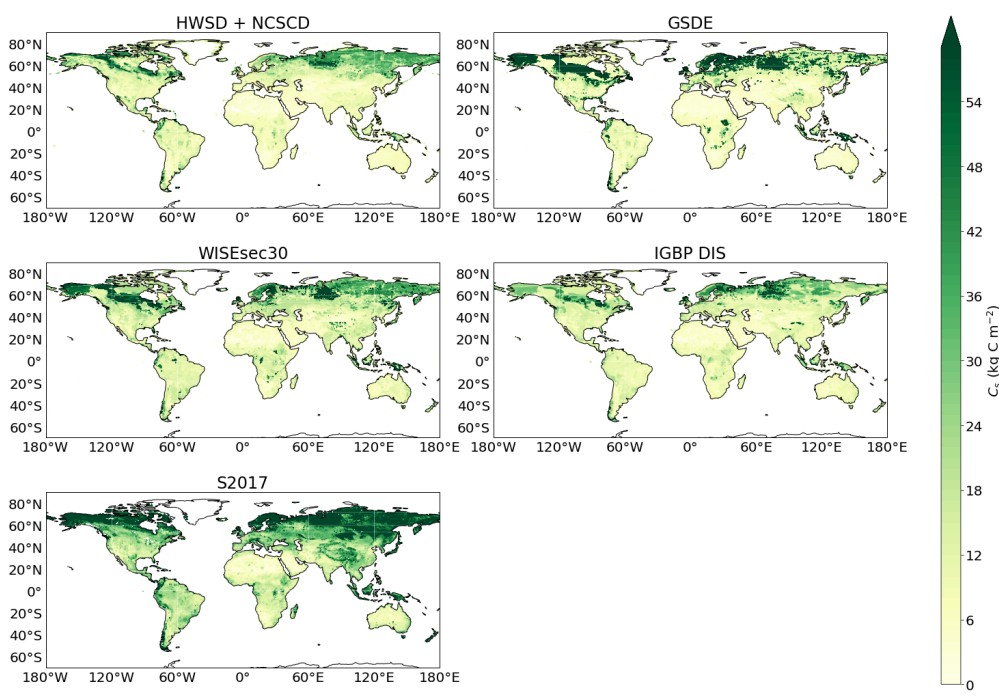

**Figure A1.** Maps comparing empirical datasets of soil carbon ($C_s$). The benchmark dataset is a map plot showing $C_s$ approximated to a depth of 1m by combining the Harmonized World Soils Database (HWSD) (FAO and ISRIC, 2012) and Northern Circumpolar Soil Carbon Database (NCSCD) (Hugelius et al., 2013), where NCSCD was used where overlap occurs. Additional map plots are shown for empirical $C_s$ estimated by: the World Inventory of Soil property Estimates (WISE30sec) (Batjes, 2016), the named 'S2017' from Sanderman et al. (2017), the Global Soil Dataset for use in Earth System Models (GSDE) (Shangguan et al., 2014), and the Global Gridded Surfaces of Selected Soil Characteristics (IGBP-DIS) (IGBP, 2000).



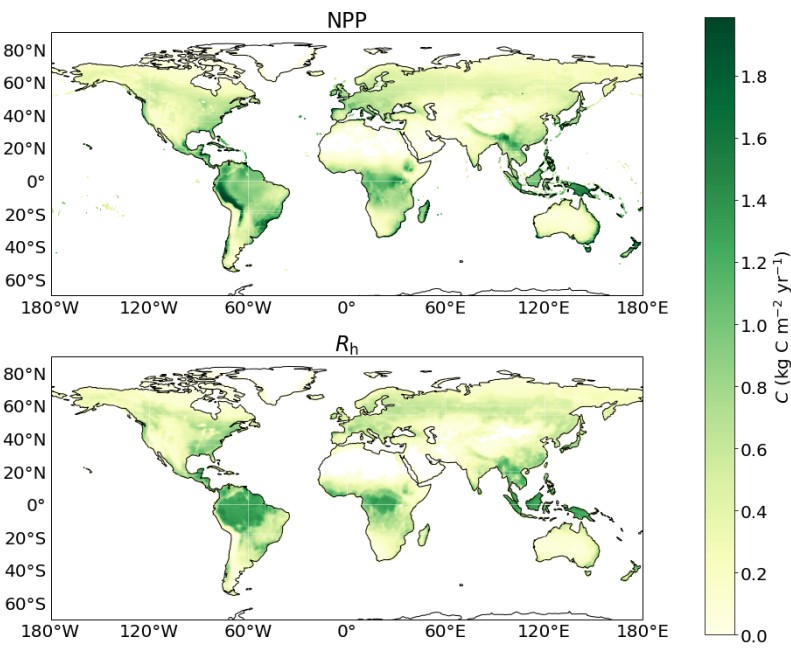

**Figure A2.** Maps of empirical carbon flux datasets. Net Primary Production (NPP) is approximated using the MODIS NPP dataset (Zhao et al., 2005), and Heterotrophic Respiration ($R_h$) is approximated using the CARDAMOM $R_h$ dataset (Bloom et al., 2015).



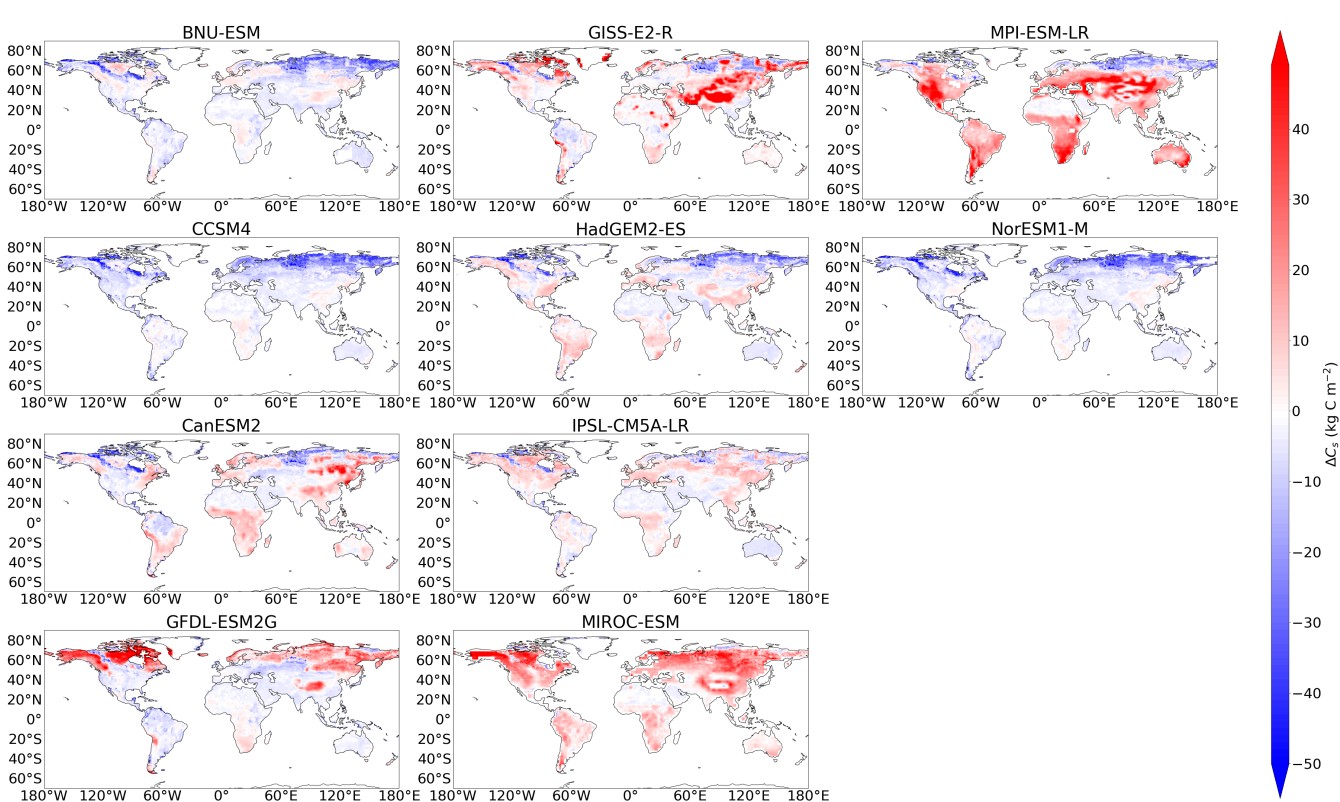

**Figure A3.** Maps of the difference in soil carbon ($C_s$) between the historical simulation (1950-2000) for the CMIP5 models and the benchmark dataset.



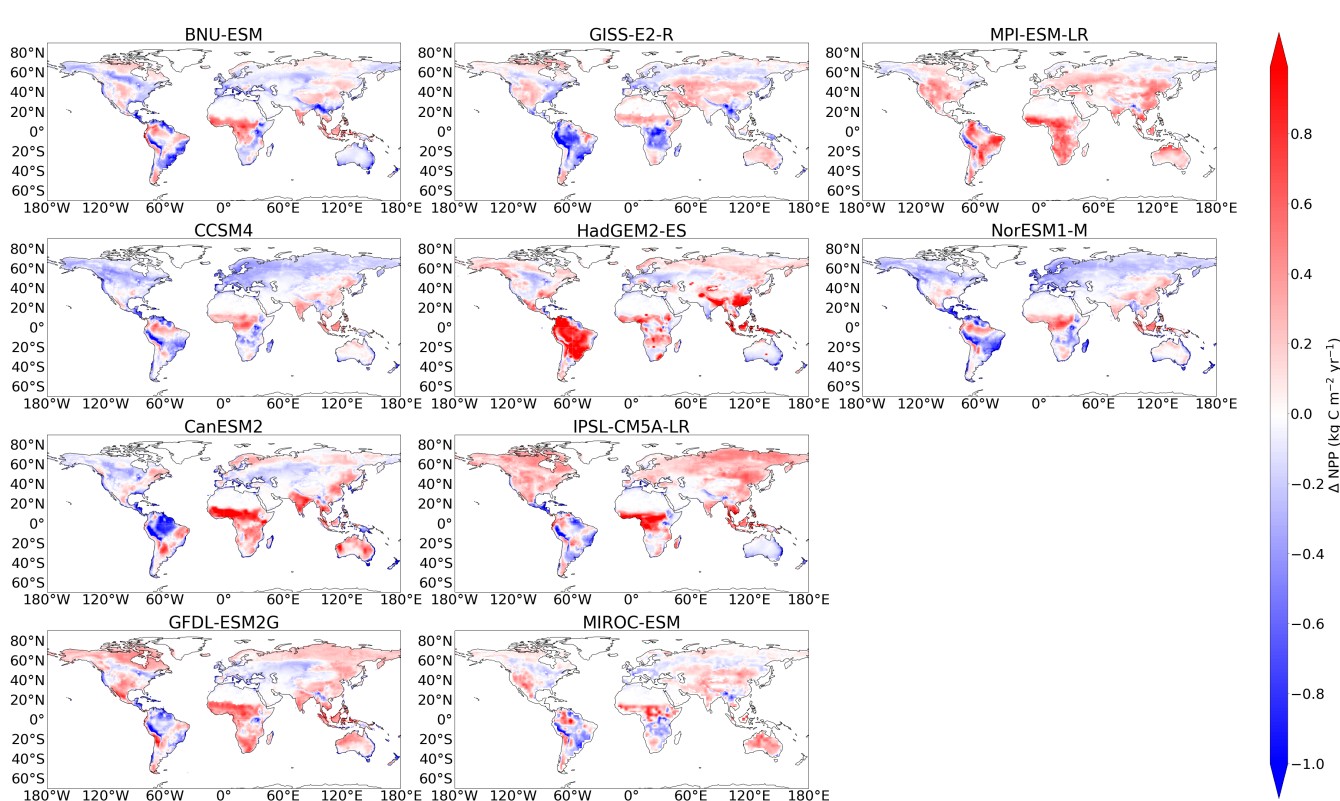

**Figure A4.** Maps of the difference in NPP between the historical simulation (1995-2005) for the CMIP5 models and the benchmark dataset.



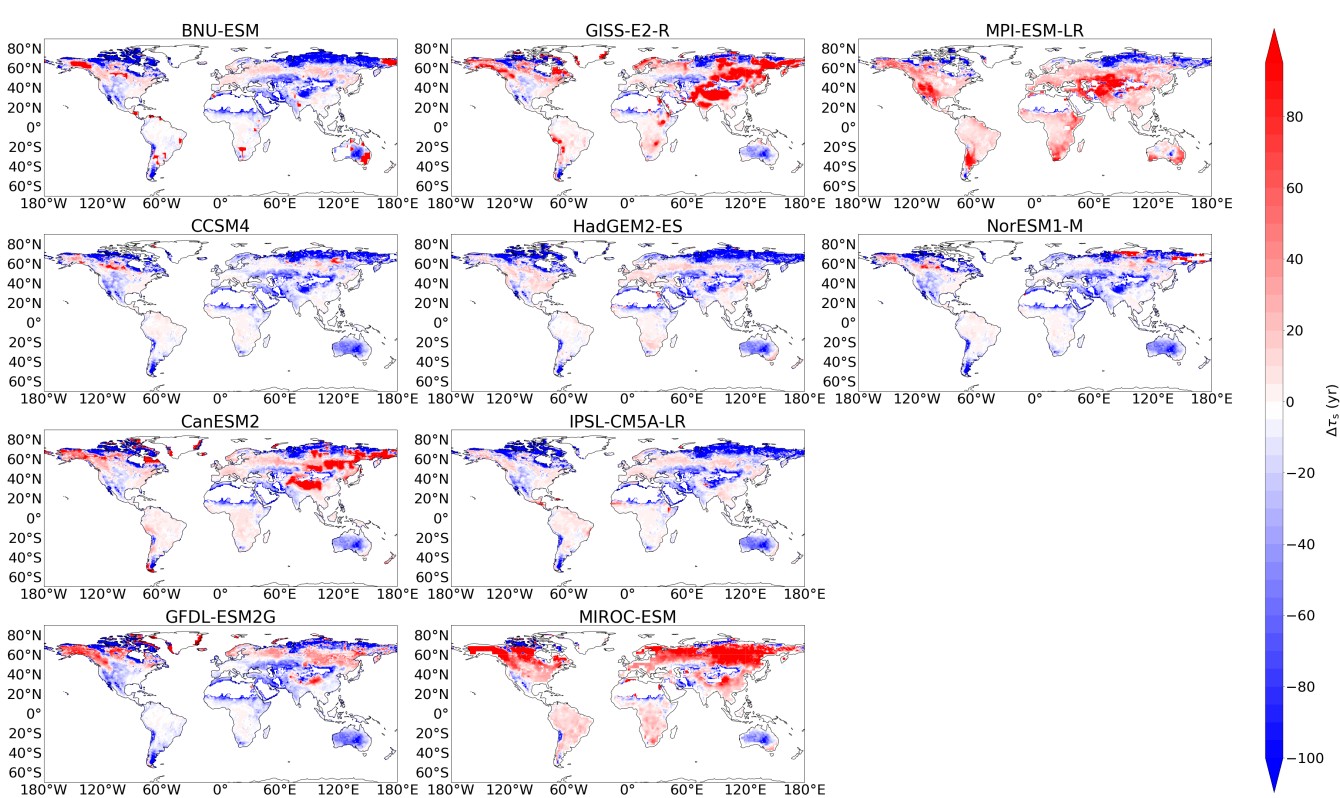

**Figure A5.** Maps of the difference in $\tau_s$ between the historical simulation for the CMIP5 models and the benchmark datasets, where $\tau_s$ is defined as the ratio of $C_s$ (1950-2000) to $R_h$ (1995-2005).





**Table A1.** Table presenting global carbon fluxes (PgC yr$^{-1}$), NPP and $R_h$, for the 11 CMIP6 models included in this study and the empirical benchmark datasets.

| Earth System Model | NPP (PgC yr$^{-1}$) | $R_h$ (PgC yr$^{-1}$) |
|---|---|---|
| ACCESS-ESM1.5 | 45.6 | 45.1 |
| BCC-CSM2-MR | 51.2 | 48.9 |
| CanESM5 | 75.5 | 75.0 |
| CESM2 | 43.9 | 38.3 |
| CNRM-ESM2-1 | 45.6 | 40.3 |
| GFDL-ESM4 | 52.6 | 43.7 |
| IPSL-CM6A-LR | 46.4 | 39.9 |
| MIROC-ES2L | 59.1 | 52.7 |
| MPI-ESM1.2-LR | 58.9 | 53.4 |
| NorESM2-LM | 43.5 | 38.2 |
| UKESM1-0-LL | 60.8 | 57.5 |
| Ensemble mean | 53.0 ± 9.39 | 48.4 ± 10.5 |
| Benchmark datasets | 56.6 ± 14.3 | 51.7 ± 21.8 |





**Table A2.** Table presenting global carbon fluxes (PgC yr$^{-1}$), NPP and $R_h$, for the 10 CMIP5 models included in this study and the empirical benchmark datasets.

| Earth System Model | NPP (PgC yr$^{-1}$) | $R_h$ (PgC yr$^{-1}$) |
|---|---|---|
| BNU-ESM | 44.3 | 42.5 |
| CCSM4 | 42.9 | 41.4 |
| CanESM2 | 59.0 | 58.8 |
| GFDL-ESM2G | 74.4 | 62.7 |
| GISS-E2-R | 31.0 | 39.5 |
| HadGEM2-ES | 69.1 | 67.0 |
| IPSL-CM5A-LR | 76.6 | 62.4 |
| MIROC-ESM | 47.1 | 41.2 |
| MPI-ESM-LR | 73.5 | 59.9 |
| NorESM1-M | 45.0 | 41.3 |
| Ensemble mean | 56.3 ± 15.4 | 52.8 ± 10.7 |
| Benchmark datasets | 56.6 ± 14.3 | 51.7 ± 21.8 |