# Peer review of "Evaluation of soil carbon simulation in CMIP6 Earth System Models"

_Biogeosciences, 2022_

## Author Comment (AC1)

Reviewer comments
Author responses

This is a useful and reasonably thorough assessment of CMIP6 Earth System Models' ability to simulate soil carbon in comparison to both CMIP5 and observations. The paper is well written, and the subject matter is entirely appropriate for Biogeosciences.

We thank the reviewer for a thorough and constructive review. We have addressed the comments given and feel the study has now been improved.

**Major comments**

Multiple conclusions are reached around CMIP6 being differently constrained in performance to CMIP5. The problem is, both CMIP5 and CMIP6 are ensembles of opportunity and you get to analyse what is there, not what you would like to be there. So, the 11 CMIP6 models are different to the 10 CMIP5 models in many ways. Some CMIP6 models are tweaks on CMIP5, some are significantly updated. There are different climate models, there are models with shared land surface models (CESM2, NorESM2-LM) and I would bet that there are shared modules within some of these land models. Obviously, you cannot compare like-with-like and I do not think there are enough members of the ensemble to suggest testing for independence. Thus, in my view all you can do is be very careful in your discussion and conclusions. For example, line 268 – is this reduction real or a function of the ensemble construction? Similar, lines 514, 580, 551 (and others) feel to me to be important in this context. You cannot "fix" this. But you can highlight it explicitly and thoughtfully in the Discussion and make sure your conclusions are limitations-aware.

As the reviewer states this issue cannot be fixed, however a conscious effort has been made throughout the manuscript to address this comment. CMIP5 and CMIP6 are ensembles of opportunities, so this point has been recognised and the non-independent nature of CMIP ensembles are now included within the Discussion:

*"Simulating global soil carbon stocks that are consistent with empirical data is required to predict reliable projections of future soil carbon storage and emission (Todd-Brown et al., 2013). Despite a reduced spread in model estimates of global total soil carbon within CMIP6 relative to CMIP5, discrepancies remain in the consistency of these estimates with the observations between the two CMIP generations.* **It should be noted that CMIP6 does not simply contain updated versions of every model in CMIP5, some new models are included and some CMIP5 models not included in CMIP6.**  **These factors** *together with the uncertainty associated with empirical datasets has resulted in no robust conclusion being drawn on the improvement of soil carbon simulation in CMIP6 compared to CMIP5. Due to the potential significant feedback that exists between soil carbon and global climate, this lack of consistency * **reduces** *our confidence in future projections of climate change (Friedlingstein et al., 2006; Gregory et al., 2009; Arora et al., 2013; Friedlingstein et al., 2014).*

*A caveat of this evaluation study is the non-independent nature of CMIP ESMs, where for example CESM2 and NorESM2-LM share the same Land Surface Model (LSM). Additionally, the ensembles included here do not necessarily represent all models that exist within each CMIP generation. However, the evaluation completed here allows for general improvements in the simulation of soil carbon stocks and fluxes between the CMIP5 and CMIP6 generations to be noted, and key areas for future model development to be highlighted.*"

Additionally, a conscious effort has been made to address this issue throughout the manuscript, where similar statements to the ones highlighted have been changed to note the non-independent nature of CMIP. Bold statements, for example "reduced uncertainty" have been replaced with "reduced model spread", and changes to the suggested lines are given below:

Line 268 – "*The ensemble mean global total soil carbon is found to have reduced in CMIP6 from CMIP5, **although it is noted that this may be a factor of the selection of models included in each ensemble rather than any change in process representation**.*"

Line 514 – "."

Line 551 – " **An improved** *simulation of NPP is*  **suggested** *in the* **ESMs included from** *CMIP6,*  *compared with the* **ESMs from** *CMIP5* . *This conclusion is* **suggested**  *by:*"

Line 580 – "*The systematic improvements*  **suggested from the evaluation** *of NPP*  **within our** *CMIP6* **ensemble** *are not*  **suggested for** *the simulation of soil carbon turnover time ($\tau_s$), where the simulation of $\tau_s$ is found to remain inconsistent with the empirical data in CMIP6* **from CMIP5**. *Improvements are*  **suggested** *within CMIP6 relative to CMIP5,*"

A more specific question is should you be showing the ensemble mean or the ensemble median? The median is more insensitive to outliers. Why show the mean?

It is true that the mean is more sensitive to outliers compared to the median, however the outlying models are not necessarily less 'realistic' than the other models. For example, CESM2 and NorESM2-LM are often outliers due to the representation of vertically resolved soil carbon. Therefore, we feel outlying models should not be given less waiting in the ensemble analysis and it is not appropriate to be insensitive to outliers with this type of evaluation. Additionally, showing the ensemble mean makes this study more comparable to other evaluation studies, as the mean value is commonly used for ESM evaluation, including within IPCC reports. Nevertheless, a map is shown below comparing the mean and median soil carbon for the considered CMIP6 ESM ensemble. This is shown to make little difference to the results, given the large spread seen within the results.

[Figure]

Abstract – there are a series of really nice results in this paper that are not to be found in the abstract. For example, line 565 might have been shown before for CMIP5 but it is important that it is still there in CMIP6. Lines 601-602 are also important.  The abstract basically says CMIP6 is an improvement – but hidden later are some really important conclusions that hint that this improvement might be related to the ensemble or at least "improvements, but for the wrong physical reasons". I think your paper will have much more impact if some of the important findings are reflected in the abstract.

The abstract has been changed to the following:

*"The response of soil carbon represents one of the key uncertainties in future climate change. The ability of Earth System Models (ESMs) to simulate present day soil carbon is therefore vital for reliably **estimating global carbon budgets required for Paris agreement targets.** In this study CMIP6 ESMs are evaluated against empirical datasets to assess the ability of each model to simulate soil carbon and related controls: Net Primary Productivity (NPP) and soil carbon turnover time ($\tau_s$). Comparing CMIP6 with CMIP5, **a lack of consistency** in modelled soil carbon remains, particularly the underestimation of northern high latitude soil carbon stocks. There is a robust improvement in the simulation of NPP in CMIP6 compared with CMIP5, however **an unrealistically high correlation to soil carbon stocks remains, suggesting the potential for an overestimation of the long-term terrestrial***

*carbon sink. Additionally, the same improvements are not seen in the simulation of $\tau_s$. These results suggest much of the uncertainty associated with modelled soil carbon stocks can be attributed to the simulation of below ground processes, and greater emphasis is required on improving the representation of below-ground soil processes in future developments of models. These improvements would help reduce the uncertainty of projected carbon release from global soils under climate change and to increase confidence in the carbon budgets associated with different levels of global warming."*

Line 154  you use mrsos. I understand why. This is soil moisture over 10 cm. You use this with 1m soil carbon. At the very least you need to discuss whether this is important to your analysis in a "caveat" section in the Discussion. In effect, you are using a high frequency soil moisture (I accept primarily for patterns) which probably reflects high frequency rainfall more than anything else. I am not proposing you change it, but the implications of this needs to be explained and discussed. Your observed soil moisture is also over a shallow depth I suspect (line 220). Does this matter – perhaps not because mostly you only consider patterns but I was less sure when you look at the constraints later.

Line 591 and whole paragraph. There is an issue hidden in here that might be worth discussing which I think was picked up in papers by Jeff Exbryat. Relations with temperature are anchored in reality – and are relatively easy. A land model should simulate a temperature that reflects observations. In contrast, soil moisture is far more complex and most land models achieve a value of soil moisture that reflects the value that is needed to constrain evaporation. There is, of course, much more to it than this, but soil moisture can vary a lot between land models, while all those land models simulate similar drainage and evaporation. This means that when looking at correlations between ts and theta, the theta is not necessarily translatable across models. So, I think there is much more to it that you hint at in this paragraph and it might be worth teasing that out a little. Specifically, the statement "is likely to be due to key soil processes not being represented" might be true but is not very insightful … and it might be more to do with the relationships between existing ley soil processes not being blended well.

The reviewer raises a good point about the discrepancy in depth between the soil moisture and soil carbon datasets, which is important to clarify. We have firstly added on line 156 (methods):

*"It is noted that this represents surface soil moisture and does not match the depth over which soil carbon is evaluated (0-1m). This is due to deeper soil moisture products not being as readily available due to limitations of remote sensing methods in penetrating deeper ground. It is expected that the surface soil moisture will be related to deeper soil moisture to some extent but will be influenced by different processes. For example, high surface soil moisture after rainfall events could run off and thus not always reach the deeper soil. However, due to the long timescales considered (1978 - 2000) for both the modelled and empirical data, the average surface soil moisture will be closely related to deeper moisture."*

In the discussion we have edited the paragraph that the reviewer highlights (from line 591) to include discussion of the caveats of the soil moisture data/simulations:

"To simulate $\tau_s$ consistently with observations, the relationship of $\tau_s$ to both temperature (T) and moisture ($\vartheta$) must also be simulated in a way that is consistent with observations. Generally, the $\tau_s$-T relationship is consistently simulated, however there is variation in the modelled temperature sensitivity of $\tau_s$ across the ensemble. The $\tau_s$-$\vartheta$ relationship is less consistently represented, where the majority of CMIP6 models do not match the empirically derived relationship. Despite a positive dependence of soil respiration on soil moisture **in the empirical data**, many of the CMIP6 models display a contradictory positive $\tau_s$-$\vartheta$ correlation (Fig. 8). This lack of consistency between the modelled and empirical relationships involving $\tau_s$ **could be influenced by a number of factors.**  Particularly, a limitation of the $\tau_s$-$\vartheta$ relationship in ESMs is the representation of peat not being simulated. Peat forms in wet areas globally, so to simulate the $\tau_s$ relationship to soil moisture consistently with empirical data, models must simulate increased, longer turnover times in regions where peat exists. Moreover, to accurately simulate the accumulation of peat in models, the soil column must be vertically resolved to allow for the soil column to grow (Chadburn et al., 2022). **However, it is noted that the empirical relationship shows $\tau_s$ reducing with higher soil moisture, which suggests that the observations are picking up more on the longer turnover times in dry areas rather than in saturated areas such as peatlands. This may be due to having only surface soil moisture information, whereas peatlands, while saturated at depth, typically have a water table ~10cm below the surface and can be very dry at the surface (Evans et al., 2021). There is also a question of what soil moisture in LSMs represents, and the definition of soil moisture varies between models. The aim within models is to act as the lower boundary condition for atmospheric models, therefore their soil parameters may historically have been tuned to give appropriate evaporation rates and not to represent the soil moisture itself.**"

Finally, I liked your conclusions, but I would like to test a couple of them. Are you sure about Conclusions 1 and 2. I mean, are you sure you can conclude this despite the uncertainties associated with the ensemble design, the observed data and so on.

Conclusions 1 and 2 have been edited to account for the associated uncertainties.

1. "The spatial patterns of soil carbon in CMIP6 models  **appear to be** more in agreement with each other than they were in CMIP5 and more consistent with observations in the mid-latitudes**, although caveats around the uncertainty in observations and the ensemble design make this conclusion uncertain.** However, soil carbon is still heavily underestimated in high northern latitudes (with the exception of two CMIP6 models that represent deep soil carbon)."

2. "Overall**, we are not able to identify**  significant improvements  in the simulation of the observed spatial pattern of soil carbon across the globe from the CMIP5 to the CMIP6 generation."

I am therefore going to recommend major revisions but with the note that they are not "major" in the sense that a lot of work needs to be done but I do suggest that resolving 1-5 would substantially improve the paper. I would suggest:

1. Add a caveats section to the discussion and focus on the major limitations of your study.
   See above comments that address the caveats of the study (Major comment numbers 1, 4, 5).
2. Check your conclusions and see if there are any minor edits you might want to make given the caveats
   See above (Major comment number 6).
3. Check over the paper and remove anything that is superfluous (see below).
   As below all minor comments.
4. Check each section to ensure that the narrative flows.
   A conscious effort has been made to ensure the study is clear and the narrative is easy to follow.
5. Make sure the reader knows the supplementary figures exist more obviously.
   These figures are now clearly referenced in the main text, and the figures have been moved to the Appendix of the study.

**Minor comments and pedantry**

Line 12 – is this statement right - that there is 2-3 times the amount of carbon in the soil cf. the atmosphere. I would have guessed its closer to an order of magnitude. Roughly 600 Pg C in the atmosphere, 550 in vegetation, 1500 in soils and 1700 in permafrost. Please check.

The IPCC AR5 reported estimates of 600 PgC within the atmosphere, however more recent estimates are approximately 800 PgC. Nevertheless, we agree that the predicted soil carbon stocks (1500+1700 = 3200) is at the upper end of our estimate (1600-2400), though less than an order of magnitude. Therefore, the sentence has been edited accordingly:

*"Soil carbon is the Earth's largest terrestrial carbon store, with a magnitude **of at least three times** of two to three times the amount of carbon contained within the atmosphere"*

Your paragraph structure needs some attention. Line 16 starts with real world carbon storage. Half way through you switch to Earth System Models. Similar issues in section 2.3.1. You do this a lot. It makes the narrative disjointed throughout and harder to follow than it should be. A significant re-write is really required focussed on the section structure.

An effort to address this comment has been made throughout the manuscript. This has mostly involved breaking up paragraphs into multiple paragraphs which now separately focus on the real-world carbon or modelled carbon, making the narrative is easily to follow.

Line 22 – How do you know that the most up to date ESMs make up CMIP6? It is an ensemble of opportunity and not everyone who can participated in CMIP6.

Line 22 – "ensemble known as CMIP6" is mis-stated. The CMIP6 ensemble is far broader than ESMs. Just be careful with the precision of the language.

This sentence has now been changed:

"**The latest generation of the Coupled Model Inter-comparison Project (CMIP) CMIP6, includes an ensemble of ESMs, which are used in the most recent Intergovernmental Panel on Climate Change (IPCC) report (AR6).**"

Line 25 – further to pedantic comments, it is not true that the carbon cycle is fundamental to obtaining accurate future projections. It is, of course, true on long timescales, but it is not true of projections to 2050. Again, be precise in the language and the timescales you are talking about.

This sentence has now been changed:

*"The response of the carbon cycle to climate change is fundamental to obtaining accurate **global carbon budgets** , and the relationships between carbon and environmental drivers used in models help to determine this response. Therefore, representing present day carbon stores and spatial controls realistically is key  **for estimating carbon emission cuts required for Paris agreement targets**."*

Line 45 - This paragraph is very confusing. It is trying to signpost what comes but fails to do that. You can just delete it as anyone reading this who cannot find your results needs help.

This paragraph has been removed.

Line 60 – I am not certain the detail in this paragraph really helps the reader. I mean, I am not sure … consider the value and whether its essential for the reader.

Line 78 – similar comment to line 60.

The content of section 2.1 in the Methods remains however the text has been edited to be more concise.

Line 110 – why divide by 1E12?

In the calculation of global totals, units are converted from kg ($10^3$) to Pg ($10^{15}$). However, this statement has been removed to avoid confusion as unnecessary detail.

Line 129 – I do not think you define the term after NPP

NPP is now defined within the methods:

*"In an unperturbed steady-state (i.e. neglecting disturbances from land-use change, fires, insect outbreaks etc.), there is no net exchange of carbon between land and atmosphere, and Rh is equal to the NPP,* **where NPP is defined as the net carbon assimilated by plants via photosynthesis minus loss due to plant respiration."**

Line 344 – the hatching is invisible on the version of the manuscript I read

The figure has been edited to make the hatching more visible and is suitable for an online journal format.

Line 510 – the word "may" is incorrect. It does reduce our confidence.

The sentence has been edited to remove the word may:
"this lack of consistency  **reduces** our confidence in future projections of climate change"

Line 526 – this is a brave statement – adding complexity does not necessarily improve simulations.

The following sentence has been edited:
" **which aims to more consistently simulate soil carbon with the real world."**

Line 621 – present day carbon is not "vital" on all timescales. Check language.

This sentence has been edited:
*"The ability of Earth System Models (ESMs) to simulate present day soil carbon is vital to help*  **predict** *reliable* **global carbon budget estimates, which are required for Paris agreement targets** *."*

Line 625-6 should be in the acknowledgment.

This line has been moved to the acknowledgments.

Figure 1-3,6,7 – I am not sure you can change this but the ensemble average of two ensembles of opportunity do not really tell us that much. Maybe you have no choice, but I'd have liked to have seen the individual models. Now, later I find you do this in the Supplementary Figures (ok – that makes sense) but unless I missed it, you do not refer to these anywhere. At the very least, you should flag the existence of these figures in the legends to Figures 1-3,6,7 and in each section when you introduce the ensemble figure you should point to the existence of the individual models.

Thank you for pointing this out. The maps of the individual models have been moved to the Appendix, so they are now easier to find. Additionally, they are now more clearly referenced, including references in the figure captions as suggested.

Figure 1-3,6,7 – again. Take a look at Figure A3. You present the ensemble average of this figure in the main text. Is that "legitimate(as distinct from commonly done). You are averaging together large changes of opposing signs.

The individual difference maps for each of the CMIP6 ESMs are also included in the main text, alongside the ensemble mean figures. Only the individual CMIP5 maps are included within the Appendix due to the focus being on CMIP6. Additionally, despite showing ensemble means, the differences between the ESMs are highlighted within the results so the opposing signs are highlighted.

Figure 4 – Ahhh, Taylor diagrams. Do you think your audience will understand them? In my experience, the only people who really understand them are those who have personally created and analysed them. For this journal I would suggest you consider their suitability.

Taylor diagrams are useful tools to present statistical information, and have been regularly used in ESM CMIP evaluation studies, so we feel they are understood. Additionally, the results from the figure are clearly explained using the diagram, which should be easy to follow even if the reader has not come across this style of diagram before.

Figure 9 – there are two shading intensities I think but its almost invisible on my printout.

The main point of this figure is to show the latitudinal pattern, shown by the solid lines, however the shading has been slightly darkened to make more visible. Therefore, the figure is suitable for an online journal format.

---

## Author Comment (AC2)

Reviewer comments
Author responses

The paper devoted to analysis of quality of terrestrial carbon cycle simulation using Earth System Models from CMIP6. Improvements of CMIP6 models comparing to CMIP5 and empirical datasets are shown. Data compared using a set of statistical parameters and colorful maps. Methods and the aim of the paper are clear.

We thank the reviewer for their comments on our study.

**Specific comments**

Soil carbon storage, net primary productivity and carbon turnover time were selected as variables responsible for terrestrial soil carbon estimations. According to suggestions NPP related with soil carbon through plant ang root litter (line 30-35), but empirical datasets have negligible correlation between these values (line 458). Pleas, give more attention for the support of your idea on relations of soil carbon and NPP.

We do not *a priori* assume a relationship between soil carbon and NPP, but we do see such a relationship clearly in the CMIP5 and CMIP6 models. Instead, we follow previous studies (Todd-Brown et al., 2013; Koven et al., 2015) in defining an effective turnover time $\tau_s$ that ensures that the soil carbon $C_s = R_h \tau_s$ at all times. For the multiannual means considered in this paper, $R_h$ is approximately equal to NPP (because the difference between NPP and $R_h$, which represents the Net Ecosystem Productivity, is a small fraction of the NPP). We can therefore safely assume that $Cs \sim NPP \tau_s$, which allows us to separate above ground drivers of soil carbon (NPP) from below-ground processes ($\tau_s$). Our analysis makes no other prior assumptions about the extent to which soil carbon is determined by NPP in the models or the observations. We make this clearer in our revised paper by editing the following text (lines 128 to 135):

*"**The definition of the effective turnover time $\tau_s = C_s / R_h$ ensures that the soil carbon at any one time is given by: $C_s = R_h \tau_s$.** In an unperturbed steady-state (i.e., neglecting disturbances from land-use change, fires, insect outbreaks etc.), there is no net exchange of carbon between land and atmosphere, and **therefore** $R_h$ is equal to **litterfall, known as fallen organic material from plants. When vegetation and soil carbon are close to a steady state, litterfall and $R_h$ are also approximately equal to** Net Primary Productivity (NPP), **where NPP is defined as the net carbon assimilated by plants via photosynthesis minus loss due to plant respiration.** In the contemporary period **considered in this study, $R_h$ has been found to be well approximated by NPP (Varney et al., 2020). This is because the difference between NPP and $R_h$, which represents the Net Ecosystem Productivity (NEP), is a small fraction of the NPP over the historical period (NPP ~ 60 PgC yr$^{-1}$; NEP ~ 3 PgC yr$^{-1}$).** Therefore, the present day soil carbon can be approximated by:*
*$C_s \sim NPP \, \tau_s$,*

*to a good accuracy. This allows for a clean separation of soil carbon variation into the above (NPP) and below ($\tau_s$) ground drivers of soil carbon spatial patterns, following the approach of previous published studies (Todd-Brown et al., 2013; Koven et al., 2015)."*

We have also edited the sentence in the Discussion which addresses this issue (line 568):

*"Despite NPP driving the spatial pattern of soil carbon stocks due to carbon input from vegetation, a positive correlation  was not expected in the real world due to regions with high soil carbon not correlating with regions of high NPP. For example, in the observational derived data soil carbon stocks are greatest in the northern latitudes due to long turnover times in these regions, whereas NPP is lower due to cold temperatures in these regions limiting vegetation growth."*

Carbon turnover time determined as a ratio of carbon amount and heterotrophic respiration. According to presented results soil carbon estimations were improved in CMIP6 comparing CMIP5, but soul carbon turnover time estimations is not good enough. Likely the issue is related with heterotrophic respiration. Could you check the hypothesis and present an analysis of quality of HR simulations?

For the reasons outlined in the previous response, heterotrophic respiration and NPP are very similar on the multiannual timescales considered in this paper. This was shown in our previous paper (Varney et al., 2020), which is now cited in the new text shown above. It is also noted that global total values for heterotrophic respiration (Rh) are presented for both CMIP5 and CMIP6 ESMs, including comparisons with observation, in the Appendix Tables A1 and A2.

Changes in soil carbon storage occurs through changes in fluxes. The accuracy of simulation of carbon fluxes will result in total estimations of soil carbon. You have shown only one flux (NPP) not directly related with soil system and give a complex parameter related with heterotrophic respiration. Is it possible to demonstrate the quality of simulations of carbon fluxes relates with soil system (i.e. heterotrophic respiration, ecosystem respiration, dissolved carbon runoff, decay rate, litterfall, etc)

The global CMIP5 and CMIP6 Earth Systems Models do not yet routinely include dissolved organic carbon (DOC). In any case, reliable global datasets of DOC are not available for model evaluation. Additionally, DOC is known to be relatively small (0.28 ± 0.07 PgC yr$^{-1}$) compared to the magnitude of NPP (approximately 60 PgC yr$^{-1}$) on a global scale considered in this study (Nakhavali et al., 2020). As explained above, NPP is a key driver of soil carbon as it provides the litterfall input, and we do have access to global datasets of NPP. Fortunately, NPP, litterfall and heterotrophic respiration are all very similar on the multiannual timescales considered in this paper (because vegetation and soil carbon are close to a steady state on those timescales). We make this clearer by adding the following sentence between lines 128 and 135 (also included above):

*"In an unperturbed steady-state (i.e., neglecting disturbances from land-use change, fires, insect outbreaks etc.), there is no net exchange of carbon between land and atmosphere, and **therefore** $R_h$ is equal to* **litterfall, known as fallen organic material from plants. When vegetation and soil carbon are close to a steady state, litterfall and $R_h$ are also approximately equal to** *Net Primary Productivity (NPP),* **where NPP is defined as the net carbon assimilated by plants via photosynthesis minus loss due to plant respiration."**

The paper contains a lot of statistical information about comparison of results from CMIP6/5 ESMs. Total estimations and spatial variability of parameters are shown. But the meaning of obtained estimations and relations with land ecosystem is missed. In the present form the paper is more suitable for Geoscientific Model Development journal where ESM and their characteristics are discussed. Understanding of reasons of ESM errors requires identification of an ecosystem types where highest discrepancies observed. Clear, that highest soil carbon is typical for peatlands. Proper simulation of peatland water, thermal and nutrient regime will giver more impact to the global carbon estimations than for other ecosystems. I suggest to emphasize the role of ecosystems in soil carbon formation and discuss the errors and improvements of ESMs not only at global scale but at ecosystem scale too.

We maintain that this study is very appropriate for publication in Biogeosciences, as it relates to previous studies in this journal (e.g., Todd-Brown et al. 2013, *Causes of variation in soil carbon simulations from CMIP5 Earth system models and comparison with observations*), and is clearly relevant to biogeochemical cycling. We note that Reviewer 1 also suggests that our paper is a good fit to the journal.

Additionally, we follow the reviewer's suggestion to discuss ecosystem types associated with the representation of soil processes, where we have added the following text to section 4.2.2 (line 601) on the important role of peatlands:

**"Different processes control soil carbon formation in different ecosystems, including stabilisation by clay particles, transformation by microbes, nitrogen and phosphorous availability, etc. (Witzgall et al. 2021). In the present study, the largest discrepancies in both soil carbon and turnover times are seen in permafrost and peatland areas (see Fig. 2 and Fig. 7). For example, the west Siberian peatland complex stands out on the majority of the panels in these figures as an area of high model error. This is partly because the soil carbon turnover times and quantities are largest in these regions, but also partly due to the specific controlling processes in these ecosystems. A key part of soil carbon development in permafrost regions is the fact that organic material can be preserved in frozen soil, including via cryoturbation and yedoma deposits, which have not yet been thoroughly represented in models (Beer, 2016; Zhu et al., 2016). There are a variety of other factors, such as plants storing significantly more of their carbon below ground instead of above ground in cold climates, and recalcitrant vegetation such as mosses, which are not represented in most ESMs (Sulman et al., 2021). Peatland formation is controlled primarily by waterlogging, which reduces oxygen available for decomposition, but there are a huge number of additional physical and biogeochemical feedbacks that take place (Waddington et al. 2015). These kinds of small-scale processes and inhomogeneities are difficult to resolve in global models with ~100km2 grid cells, and this**

*should be weighed up against their relative impact on global carbon budgets when considering including these processes in ESMs. However, it is suggested that the large-scale discrepancies such as in the permafrost and large peatland areas can and should be resolved in future model versions."*

---

## Referee Report (RR1)

First, I think the authors have revised this paper well. I am satisfied by their responses to my queries – for example the demonstration that the choice of mean or median is not material. The abstract is definitely better. I am also satisfied how they have modified the manuscript. My original review was "major revisions" with the caveat that the changes were important, but unlikely to be overly dramatic in terms of changes in the text. The changes reflected in the manuscript are more than I anticipated but welcome.

Perhaps most critically, I think the authors have edited this manuscript well, it flows more elegantly, the sentence structure is definitely better, the discussion is clearer and I feel the conclusions are now appropriately stated.

I was amused by the response to my comments about the use of *mrsos*. The statement:

However, due to the long timescales considered (1978 - 2000) for both the modelled and empirical data, the average surface soil moisture will be closely related to deeper moisture.

Prove it ☺ Or, delete this sentence. I do appreciate the additional text (except this last sentence) – I think it is useful, and just as useful if you omit this final sentence whixh will be true in many regions, but not in some that are particularly interesting. This is simply a limit to your study, a limit that could not really be avoided. Just be up-front about it.

The statement:

***There is also a question of what soil moisture in LSMs represents, and the definition of soil moisture varies between models.***

I fully agree - the famous paper about this, which you could cite, is:

- Koster, R.D., Z. Guo, R. Yang, P.A. Dirmeyer, K. Mitchell, M. J. Puma, 2009: On the Nature of Soil Moisture in Land Surface Models. *J. Climate*, **22**, 4322–4335. doi: 10.1175/2009JCLI2832.1

I'd re-state my previous comments that this paper is well-suited to this journal. It is not a model development paper and so I respectfully disagree with Reviewer 2. For the record, I read the authors responses to reviewer 2. There places where the authors effectively say "we cannot do that because of CMIP protocols" around data reported and so on. I did not see anything in these responses that I disagreed with.

In summary, I think this is a  thorough revision that accommodates my earlier comments.

---

## Author Response (AR2)

We thank the reviewer again for reviewing this manuscript, and the positive comments in this review.

The following sentence has been removed.

However, due to the long timescales considered (1978 - 2000) for both the modelled and empirical data, the average surface soil moisture will be closely related to deeper moisture.

The following paper is now cited.

Koster, R.D., Z. Guo, R. Yang, P.A. Dirmeyer, K. Mitchell, M. J. Puma, 2009: On the Nature of Soil Moisture in Land Surface Models. *J. Climate*, **22**, 4322–4335. doi: 10.1175/2009JCLI2832.1